# Can Large Language Models Master Complex Card Games?

**Wei Wang**[12], **Fuqing Bie**[3], **Junzhe Chen**[2], **Dan Zhang**[2]
**Shiyu Huang**[4], **Evgeny Kharlamov**[5], **Jie Tang**[2*]
[1]Nankai University, [2]Tsinghua University, [3]Beijing University of Posts and Telecommunications
[4]Zhipu AI, [5]Bosch Center for Artificial Intelligence
weiwangorg@163.com

## Abstract

Complex games have long been an important benchmark for testing the progress of artificial intelligence algorithms. AlphaGo, AlphaZero, and MuZero have defeated top human players in Go and Chess, garnering widespread societal attention towards artificial intelligence. Concurrently, large language models (LLMs) have exhibited remarkable capabilities across various tasks, raising the question of whether LLMs can achieve similar success in complex games. In this paper, we explore the potential of LLMs in mastering complex card games. We systematically assess the learning capabilities of LLMs across eight diverse card games, evaluating the impact of fine-tuning on high-quality gameplay data, and examining the models' ability to retain general capabilities while mastering these games. Our findings indicate that: (1) LLMs can approach the performance of strong game AIs through supervised fine-tuning on high-quality data, (2) LLMs can achieve a certain level of proficiency in multiple complex card games simultaneously, with performance augmentation for games with similar rules and conflicts for dissimilar ones, and (3) LLMs experience a decline in general capabilities when mastering complex games, but this decline can be mitigated by integrating a certain amount of general instruction data. The evaluation results demonstrate strong learning ability and versatility of LLMs. The code is available at `https://github.com/THUDM/LLM4CardGame`

## 1 Introduction

A long-term goal of artificial intelligence is to achieve superhuman performance in highly challenging domains [1–3]. Games, particularly complex ones such as Chess and Go, have become the best testing grounds for artificial intelligence algorithms [4–7]. In recent years, artificial intelligence algorithms have made significant breakthroughs in the realm of games. AlphaGo is the first to defeat human professional players in Go by using supervised learning from expert human data and reinforcement learning [4]. Following this, a general reinforcement learning algorithm, AlphaZero, achieves superhuman performance in three challenging games: Chess, Shogi (Japanese chess), and Go [6]. MuZero even achieves performance equivalent to AlphaZero without needing to know the rules of the game [7].

Recently, large language models (LLMs) [8–11] have achieved remarkable performance, even surpassing human levels, across a wide range of tasks including general knowledge question answering [12, 13], mathematics [14, 15], coding [16, 17], and agent [18, 19]. This naturally raises the question: can language models achieve superhuman performance in complex games, or at least reach the same level as the best reinforcement learning algorithms? In this paper, we focus on card games.

---

[*]Corresponding author.

39th Conference on Neural Information Processing Systems (NeurIPS 2025).

Currently, there is a vast amount of research evaluating LLMs across various tasks. Among these studies, some research also evaluates the decision-making capability of LLMs in both board games and card games, such as Texas Hold'em [20, 21], Blackjack [22], and Guandan [23, 24]. However, these studies still have some limitations. First, many evaluation studies assess LLMs through prompting without involving fine-tuning [25, 26]. These prompt-based evaluation studies can only assess whether LLMs are capable of applying their existing knowledge to new environments. *But, they do not evaluate the learning ability of LLMs.* Second, some evaluation studies include assessments of LLMs after fine-tuning and demonstrate that fine-tuning improves the performance of LLMs in new environments. *But, the tasks evaluated in these studies lack sufficient complexity, making them inadequate to comprehensively assess the learning capabilities of LLMs.*

As mentioned at the beginning, complex games are often used to explore the upper bound of artificial intelligence algorithms' learning capabilities. Therefore, in this paper, we investigate whether language models can master complex card games. To address the shortcomings of previous work, we systematically evaluate the performance of language models on eight carefully selected card games. First, most games exhibit a high level of complexity as shown in Figure 1. The high complexity of these games presents a greater challenge to the learning abilities of large models. Evaluations across the eight games provide a more comprehensive understanding of LLMs. Second, we evaluate the performance ceiling that LLMs can achieve by fine-tuning the model on high-quality gameplay interaction data. Compared to prompt-based methods, fine-tuning methods focus on evaluat-

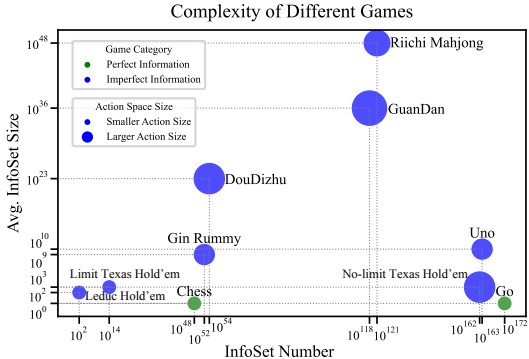

Figure 1: Complexity of the games. InfoSet Number: the number of the information sets; Avg. InfoSet Size: the average number of states in a single information set.

ing the learning ability of language models. We generate high-quality gameplay interaction data using strong game AIs or directly utilize publicly available high-quality interaction data. Specifically, we focus on the following three research questions:

**1**. *Can LLMs master complex card games? And how much data is required to master these games?*

**2**. *Can LLMs simultaneously master multiple games? Do different games mutually enhance each other or do conflicts arise between them?*

**3**. *Can LLMs maintain their general capabilities while mastering complex games?*

To answer these questions, we first fine-tune language models on each of the eight games separately to evaluate the extent to which the models can master individual games. Next, we fine-tune the models on a mixture of all the game data to assess their ability to master all the games simultaneously. Finally, we evaluate whether the models' general capabilities decline using MMLU-Pro [12], Math-500 [27], and HumanEval [28] benchmarks for knowledge question answering, math, and coding skills. Additionally, we analyze the performance variations of language models with different parameter sizes and types (Qwen2.5 [29], Llama3.1 [30], and GLM4 [9]). In summary, the contributions of this work are:

- We are the first to propose a comprehensive evaluation of the learning capabilities of LLMs across multiple high-complexity games, which present greater challenges to the learning abilities of LLMs.

- We obtain a large amount of high-quality data for LLMs to learn by utilizing strong game AIs and game prompt templates, avoiding the problem of high computational resource consumption when LLMs learn from scratch in the environment.

- We thoroughly assess the learning ability of the models through experiments in three aspects. The evaluation results demonstrate strong learning ability and versatility of LLMs, as they can simultaneously master multiple complex games without altering the model structure.

## 2 Related Works

The integration of LLM-based agents into card and board game has garnered significant attention, focusing on enhancing strategic reasoning and adaptability [31–33]. Previous research has mainly focused on improving LLM-based agents in card games with incomplete information and adversarial settings, such as Texas Hold'em [20–22, 34], Blackjack [22], and Guandan [23, 24]. For example, Yim et al. [23] introduced a Theory of Mind (ToM) planning technique for Guandan, enabling LLM agents to adapt strategies based on game rules, state, and history. Their approach also integrates an external tool to manage the game's large action space. While effective in a single game, the generalizability of such prompt-based strategies across multiple games remains an open question. Guo et al. [21] proposed Suspicion-Agent, a prompt-based approach that leverages GPT-4's reasoning and high-order ToM capabilities to adapt strategies in imperfect information card games. It enables GPT-4 to play against different opponents using only game rules and observations as input. Their experiments demonstrate its effectiveness across multiple games. However, as a prompt-based method, it relies solely on the model's inherent knowledge, which limits its overall performance.

## 3 Method

From AlphaGo, to AlphaZero, and then to MuZero, we can see that these methods have achieved significant breakthroughs in complex games by continuously exploring the environment and leveraging successful experiences. This paper aims to explore whether LLMs can master complex games similarly. Considering the time and resource consumption involved in the exploration process, we utilize existing strong game AIs to generate high-quality trajectory data. This study investigates whether LLMs can master complex games by learning from this high-quality trajectory data. Next, we introduce the selected games and the process of generating training data for each game.

### 3.1 Games

For game selection, we primarily consider the popularity, complexity, and availability of high-quality models or data. Based on these three aspects, we select eight games: DouDizhu, Guandan, Riichi Mahjong, Uno, Gin Rummy, Leduc Hold'em, Limit Texas Hold'em, and No-Limit Texas Hold'em.

**DouDiZhu.** DouDizhu[1] (a.k.a. Fighting the Landlord) is the most popular card game in China. The game is played by three players with a standard 54-card deck. There are two roles in the game: a landlord and two farmers. Some studies have explored building strong DouDizhu AIs using techniques such as handcrafted heuristic rules, reinforcement learning based methods, and search algorithms [35–37]. Among these methods, DouZero [37] is a simple and effective approach that requires no human knowledge or state/action abstraction. It is currently the strongest publicly accessible DouDizhu AI.

**GuanDan.** Guandan[2] is another popular card game in China. The game requires four players, with the two players sitting opposite each other forming a team. Its gameplay is similar to Dou Dizhu. However, Guandan is more complex in comparison. There is relatively less research on Guandan AI [38, 39]. Among the existing work, DanZero [38], which employs a neural network-enhanced Monte-Carlo method, has outperformed other algorithms. Therefore, we choose DanZero as our teacher model.

**Mahjong.** Mahjong[3] is a widely popular multiplayer tile-based game across the world. Mahjong has many variants, and this paper focuses on Riichi Mahjong (a.k.a. Japanese Mahjong). Suphx is the first Mahjong AI to defeat most top human players [40]. Then LuckJ developed by Tencent reached 10 Dan on Tenhou[4] and surpasses all human players and other AIs. However, the model weights for both of these AIs have not been made publicly available. Nevertheless, Tenhou provides gameplay data from expert players.

---

[1]https://en.wikipedia.org/wiki/Dou_dizhu
[2]https://en.wikipedia.org/wiki/Guandan
[3]https://en.wikipedia.org/wiki/Mahjong
[4]https://tenhou.net/

| Game | # Players | # Teams | Teacher model/Data | Opponent | # Games | Avg. Steps per Game | Avg. Steps per Player per Game | Total Steps (Filtered) | Avg. Legal Actions per Step | Training data |
|---|---|---|---|---|---|---|---|---|---|---|
| DouDizhu | 3 | 2 | DouZero | Rule model | 200k | 37.31 | 12.44 | 2,950k | 10.06 | 1,000k |
| GuanDan | 4 | 2 | DanZero | Rule model | 6k | 311.25 | 155.63 | 1,220k | 48.67 | 1,000k |
| Riichi Mahjong | 4 | 4 | Data from experts | Mortal | 7k | 656.92 | 164.23 | 1,090k | 8.79 | 1,000k |
| Uno | 2 | 2 | Rule model | Random | 50k | 42.33 | 21.16 | 410k | 3.14 | 400k |
| Gin Rummy | 2 | 2 | Rule model | Random | 50k | 52.14 | 26.07 | 1,280k | 6.22 | 400k |
| Leduc Hold'em | 2 | 2 | DQN model | Random | 400k | 3.61 | 1.81 | 580k | 2.86 | 400k |
| Limit Texas Hold'em | 2 | 2 | DQN model | Random | 200k | 5.01 | 2.50 | 450k | 2.96 | 400k |
| No-limit Texas Hold'em | 2 | 2 | DQN model | Random | 400k | 3.78 | 1.89 | 700k | 4.31 | 400k |

Table 1: Data generation information of games.

**Uno.** Uno[5] is a proprietary American shedding-type card game. The game is played with a specially designed 108-card deck. There are 2 players in the game. Each player starts with seven cards dealt face down. Players take turns matching the card in the Discard Pile by number, color, or symbol/action. The objective is to be the first player to get rid of all the cards in hand.

**Gin Rummy.** Gin Rummy[6] is a two-player card game. The game is played by two players using a standard 52-card deck. The dealer deals 11 cards to the opponent and 10 cards to himself. During each turn, you can pick up the discard or draw from the face-down stock, then discard a card. Players try to form melds of 3 or more cards of the same rank or 3 or more cards of the same suit in sequence.

**Leduc Hold'em.** Leduc Hold'em, introduced in Southey et al. [41], is a simplified variant of Limit Texas Hold'em. This version uses a deck comprising only six cards, with two pairs each of King, Queen, and Jack. The game involves two players, spans two rounds, and has a maximum of two bets per round, with the raise amounts fixed at 2 in the first round and 4 in the second round.

**Limit Texas Hold'em.** Limit Texas Hold'em[7] is a well-known betting game with 52-card deck. Each player receives two face-down cards, known as hole cards. Subsequently, five community cards are dealt in three stages: the flop, the turn, and the river. Players aim to form the best possible five-card hand using any combination of their hole cards and the community cards.

**No-limit Texas Hold'em.** No-limit Texas Hold'em follows similar rules to Limit Texas Hold'em but with key differences in betting. No-limit Texas Hold'em allows players to raise by at least the amount of the previous raise in the same round and up to the entirety of their remaining stack. Additionally, there is no limit on the number of raises in No-limit Texas Hold'em.

### 3.2 Data Preparation

**Trajectory Generation.** We generate gameplay interaction data by having the teacher model compete against opponents. The teacher model and opponent information for each game are shown in Table 1. For DouDizhu, we use DouZero [37] as the teacher model and a rule-based model [42] as the opponent model. For GuanDan, we use DanZero [38] as the teacher model and a rule-based [38] model as the opponent model. For Riichi Mahjong, we download the match data of human professional players from the Tenhou[8] platform for the year 2020. The opponent model is Mortal[9], a strong Mahjong AI, which is used only during evaluation. For Uno and Gin Rummy, we use rule model from Zha et al. [42] as the teacher model and use random as the opponent. For Leduc Hold'em, Limit Texas Hold'em, No-limit Texas Hold'em, we train DQN model as the teacher model with RLCard framework[10].

Based on the complexity of different games, we play each game a varying number of times. The important information of the generated data is shown in Table 1. From the table, it can be seen that the average number of steps in the games Doudizhu, Guandan, and Mahjong is significantly higher than the other games. Particularly, Guandan and Mahjong have longer steps because each game consists of multiple rounds. For example, in Guandan, the game progresses from 2 to Ace.

**Trajectory Filtering.** In this paper, we consider each observation-action pair of one step as a sample. We filter the generated data based on two criteria to obtain high-quality data. First, we only retain the

---

[5]https://en.wikipedia.org/wiki/Uno_(card_game)

[6]https://en.wikipedia.org/wiki/Gin_rummy

[7]https://en.wikipedia.org/wiki/Texas_hold_%27em

[8]https://tenhou.net/

[9]https://github.com/Equim-chan/Mortal

[10]https://github.com/datamllab/rlcard

observation and action data of the winning player. Additionally, we consider each observation-action pair from the player as a single data instance. Second, for all eight games, the environment provides the legal action options per action, and we only retain data samples where the number of legal action options is greater than one. The amount of data obtained after filtering is presented in Table 1. It can also be seen from the table that the number of legal candidate actions per sample for Doudizhu, Guandan, and Mahjong exceeds that of the other five games, making these three games relatively more complex.

**Supervised Fine-Tuning Data Generation.** To perform instruction fine-tuning on the model, we design prompts for each game to convert observation-action pairs into instructions and corresponding outputs. The instruction primarily consists of three parts: game introduction, state data, and output format instructions. The game introduction includes the game rules and the player's objectives. The state data comprises information such as the player's hand, community cards, the sequence of historical actions, and legal actions. The output format specifies that the model should output actions in JSON format. Complete instructions for each game can be found in Appendix A.3.

# 4 Experiments

## 4.1 Experiment Setup

**Data.** Through the data synthesis process described in Section 3.2, we obtain the training data for each game. For DouDizhu, GuanDan, and Riichi Mahjong, we sample 1,000k instances as training data. For Uno, Gin Rummy, Leduc Hold'em, Limit Texas Hold'em, and No-limit Texas Hold'em, we sample 400k instances as training data.

**Model.** We fine-tune three different types of language models—Qwen2.5-7B-Instruct [29], Llama3.1-8B-Instruct [30], and GLM4-9B-Chat [9]—to analyze the impact of model type on performance. Additionally, we fine-tune Qwen2.5 models with different parameter sizes, ranging from 0.5B to 14B parameters, to evaluate the effect of model size on performance. We fine-tune all models with LLaMA-Factory Framework [43] and use LoRA fine-tuning [44]. The LoRA rank and LoRA alpha are set to 8 and 16, respectively. We fine-tune all models with 1 epoch. We apply a peak of 1e-4 learning rate with a cosine scheduler. The batch size is 128. We conduct experiments on a server with 8 H100 GPUs.

**Metric.** For DouDizhu, we use the win rate. For GuanDan, we use the round win rate. For Riichi Mahjong, Uno and Gin Rummy, Leduc Hold'em, Limit Texas Hold'em, No-limit Texas Hold'em, we use reward score. The reward score in Mahjong is determined based on the average rank over multiple games, with the rewards for ranks 1, 2, 3, and 4 being 3, 2, 1, and 0, respectively. The reward scores for the other five games can be found in the RLCard framework[11]. We evaluate the LLM by having it play multiple games against opponents. The number of games for the eight games are 1000, 20, 50, 500, 100, 1000, 1000, and 1000, respectively. For DouDizhu, following DouZero [37], we have the LLM play 500 times as the Landlord and 500 times as the Farmers, then calculate the average win rate for both roles.

## 4.2 RQ1. *Can LLMs master complex card games? And how much data is required for them to master these games?*

**Experimental Design.** We fine-tune the language model separately on each game's data and then evaluate its performance on the respective game. The training data information for each game is presented in Table 1. For each game, we train on all training data for one epoch, saving a checkpoint every 400 steps. This allows us to analyze the model's performance changes with different amounts of training data. Additionally, we fine-tune three different types of LLMs and five different sizes of LLMs to explore the impact of model type and size on performance.

**Results and Analysis.** The results of DouDizhu, GuanDan, and Mahjong are shown in Figure 2a-Figure 2c. As shown in the figure, with the increasing amount of training data, the performance of the LLM in Doudizhu and Guandan gradually approaches that of the teacher model. In Mahjong, even though there is no available teacher model, the LLM has achieved performance comparable to

---

[11]https://rlcard.org/

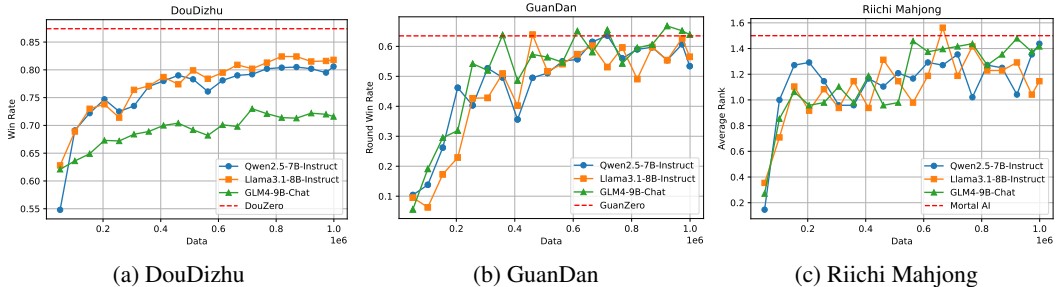

| (a) DouDizhu | (b) GuanDan | (c) Riichi Mahjong |

Figure 2: Performance of different training data.

that of a very strong Mahjong AI. Figure 1 and Table 1 have already shown that these three games have high complexity (long average decision steps and a large number of valid actions per step). These results indicate that, **given sufficient high-quality data, LLMs can master complex card games**. As training progresses, the model acquires more and more game strategic knowledge, leading to a continuous improvement in win rate. It is worth noting that DouZero actually consists of three models, with one model trained for the Landlord and two models for the two Farmers. In contrast, LLMs can play all three roles with a single model, further demonstrating their powerful learning capability and versatility.

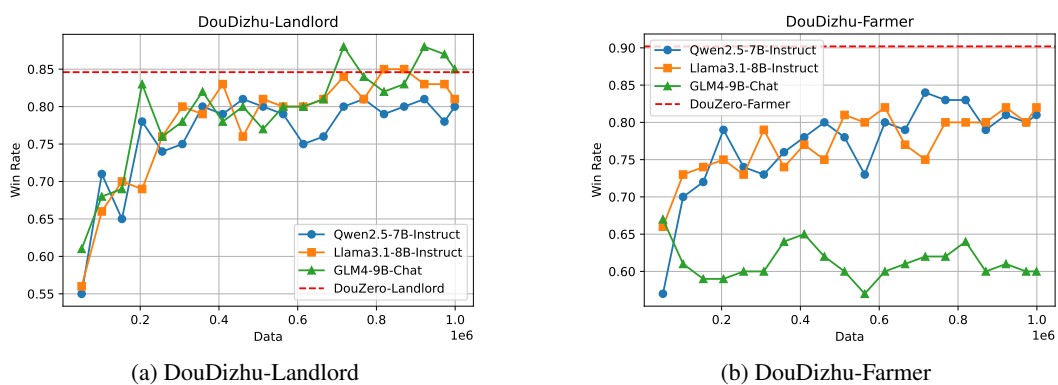

| (a) DouDizhu-Landlord | (b) DouDizhu-Farmer |

Figure 3: Performance of different roles.

**Different Games and Model Types.** As shown in Figure 2b and Figure 2c, in Guandan and Mahjong, there is no significant difference in performance among the three models, indicating that the learning capabilities of the three models are comparable. *However, in DouDizhu, the performance of GLM is significantly worse than Qwen and Llama.* To analyze the performance differences of different models in DouDizhu, we further plot the win rates of the models when playing different roles, as shown in Figure 3a and Figure 3b. As observed from Figure 3a and Figure 3b, GLM performs better than Qwen and Llama in DouDizhu-Landlord, while performing significantly worse than the two models in DouDizhu-Farmer. This suggests that GLM did not effectively balance the learning between the two roles and focused more on the landlord role, leading to weaker performance for the farmer role.

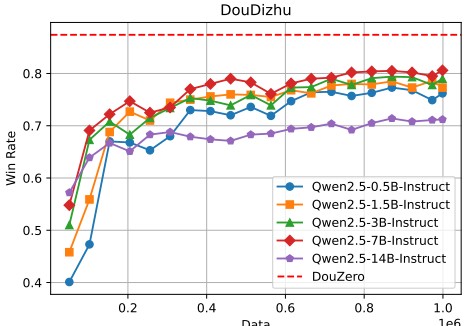

Figure 4: Performance of models with different sizes on DouDizhu.

However, comparing Figure 3a and Figure 3b, we discovered another strange phenomenon: *for DouDizhu, why is there such a difference between Landlord and Farmer performance? And why are all of the models so much lower than ceiling performance for the Farmer role?* Upon analyzing the training data, we suspect that this is caused by the filtering rules applied to the game data. In DouDizhu, there is one landlord and two farmers, with each game resulting in either a victory for the landlord or for the farmers. During filtering, we retained only the data for the winning side. When

the farmers win, the data for both farmers is retained. However, in many cases, the victory may have been primarily driven by the actions of one farmer, while the data for the other farmer may be of lower quality. Consequently, the training data includes some low-quality data for the farmer role, leading to performance for the farmer role that falls significantly below its ceiling performance.

**Different Model Sizes.** We train and evaluate five different sizes of Qwen2.5 models on Doudizhu and the results are shown in Figure 4. From 0.5B to 7B, the performance of the models gradually improves as the number of model parameters increases, indicating a positive correlation between model size and performance. However, we notice that despite the 14B model having the largest number of parameters, its performance is the worst.

| Model | Landlord | Farmer | Average |
|---|---|---|---|
| Qwen2.5-7B-Instruct | 0.828 | 0.784 | 0.806 |
| Qwen2.5-14B-Instruct | 0.858 | 0.570 | 0.714 |

Table 2: Win rate of 7B and 14B models.

In order to analysis this, we further plot the win rates of the models when playing different roles in Table 2. From the table, 14B model performs better as the landlord (approaching the performance of the teacher model) but significantly worse as the farmer. This results in the average win rate of the 14B model being lower than that of the 7B model. This is similar to why the GLM model performs worse on DouDizhu compared to Qwen and Llama.

### 4.3 RQ2. *Can LLMs simultaneously master multiple games? Do different games mutually enhance each other or do conflicts arise between them?*

**Experimental Design.** Based on the above experimental results, we roughly determine the amount of data required for each game to converge. We then sample data from the training datasets of each game according to this amount and merge them to obtain a mixed training set that includes data from all games. Specifically, the combined dataset contains 3.1 million data points, with the quantities of the eight games being: 700k, 950k, 650k, 200k, 50k, 250k, 200k, and 100k, respectively. We empirically determine the quantity of instances for each game based on game complexity and the results of Experiment 4.2. For example, games with higher complexity necessitate a larger volume of training data. We fine-tune the language model on this mixed training set to evaluate whether it can simultaneously master multiple games. We compare the fine-tuned models with the API-based models and the base models.

| Model | DouDizhu | GuanDan | Riichi | Uno | Gin Rummy | Leduc | Limit | Nolimit |
|---|---|---|---|---|---|---|---|---|
| *API-based models* | | | | | | | | |
| GPT-4o-mini | 0.195 | 0.019 | 0.15 | 0.128 | -0.176 | 0.30 | 0.45 | 2.47 |
| GPT-4o | 0.180 | 0.019 | **0.25** | 0.072 | **0.405** | 0.84 | 0.60 | 2.73 |
| GLM-4-air | 0.330 | 0.000 | 0.10 | -0.068 | -0.415 | -0.12 | **1.13** | -0.89 |
| GLM-4-plus | **0.345** | 0.019 | 0.00 | 0.020 | -0.344 | 0.80 | 0.63 | **3.21** |
| DeepSeek-V3 | 0.320 | 0.000 | 0.15 | 0.128 | 0.147 | 0.77 | 0.22 | 0.18 |
| DeepSeek-R1 | 0.185 | **0.020** | 0.05 | **0.148** | 0.228 | **0.88** | 0.24 | 1.88 |
| *Base models* | | | | | | | | |
| Qwen2.5-7B-Instruct | 0.087 | 0.000 | 0.04 | 0.032 | -0.530 | 0.63 | 1.05 | 1.25 |
| Llama3.1-8B-Instruct | 0.155 | 0.000 | 0.08 | 0.120 | -0.463 | 0.62 | -0.04 | -2.10 |
| GLM4-9B-Chat | 0.131 | 0.000 | 0.08 | 0.000 | -0.362 | 0.52 | 0.85 | -0.44 |
| *Fine-tuned models* | | | | | | | | |
| Qwen2.5-7B-Instruct-mix | 0.852 | 0.634 | 1.08 | 0.108 | 0.177 | **1.24** | 2.66 | 4.86 |
| Llama3.1-8B-Instruct-mix | 0.870 | 0.661 | **1.38** | 0.164 | 0.186 | **1.24** | 2.77 | **6.02** |
| GLM4-9B-Chat-mix | **0.882** | **0.698** | 1.31 | **0.252** | **0.191** | **1.24** | **2.89** | 5.77 |

Table 3: Performance of different models on all games. Bold font indicates the maximum value in each category, and underline indicates the second-highest value. Mix refers to models fine-tuned on the mixed training set composed of data from all games.

**Results and Analysis.** The results are shown in Table 3. All API-based models score relatively low on the two most complex games, GuanDan and Riichi, while their scores are relatively higher on the other six games. DeepSeek-R1 performed the best among all API-based models, achieving the highest scores in three games. This demonstrates the effectiveness of the reasoning mode. We observe that models of the same type with larger parameter versions or reasoning versions perform better

than those with smaller parameters or non-reasoning versions. For example, DeepSeek-R1 shows improvements in most games compared to DeepSeek-V3. GLM and DeepSeek's models score higher in DouDizhu, likely because this game is quite popular in China. Compared to the API-based models, the three base models perform worse in most of the games. Compared to the API-based models and base models, the fine-tuned model achieve the best performance in most of the games. Notably, in the complex games of Doudizhu, Guandan, and Riichi, their performance improve significantly. These results indicate that, after being trained on multiple high-quality game datasets, LLMs can simultaneously master multiple complex games.

| Model/Game | DouDizhu | GuanDan | Riichi | Uno | Gin Rummy | Leduc | Limit | Nolimit |
|---|---|---|---|---|---|---|---|---|
| DouDizhu | **0.806** | 0.010 | 0.08 | 0.032 | -0.528 | 0.637 | 1.16 | 2.54 |
| GuanDan | 0.294 | **0.636** | 0.13 | -0.004 | 0.030 | 0.637 | 1.10 | 2.62 |
| Riichi | 0.022 | 0.010 | **1.44** | 0.000 | -0.233 | 0.637 | 0.91 | -0.87 |
| Uno | 0.101 | 0.000 | 0.06 | **0.220** | 0.028 | 0.637 | 1.14 | 1.45 |
| Gin Rummy | 0.039 | 0.010 | 0.06 | 0.136 | **0.196** | 0.637 | 0.97 | -0.34 |
| Leduc | 0.082 | 0.010 | 0.10 | -0.032 | -0.584 | **1.244** | 2.56 | 7.58 |
| Limit | 0.165 | 0.019 | 0.04 | -0.008 | -0.520 | 1.176 | **2.84** | 4.83 |
| Nolimit | 0.118 | 0.000 | 0.10 | -0.056 | -0.432 | 1.012 | 2.12 | **7.75** |
| Mix | 0.852 | 0.634 | 1.08 | 0.108 | 0.177 | 1.244 | 2.66 | 4.86 |

Table 4: Influence between different games using Qwen model. Each row represents the performance of a model trained on one specific game across all games. Mix refers to models fine-tuned on the mixed training set composed of data from all games. Bold indicates the maximum value, and underline indicates the second-highest value, both excluding the mix model.

| Model/Game | DouDizhu | GuanDan | Riichi | Uno | Gin Rummy | Leduc | Limit | Nolimit |
|---|---|---|---|---|---|---|---|---|
| DouDizhu | **0.824** | 0.000 | 0.13 | 0.008 | -0.496 | 0.637 | 1.14 | 3.21 |
| GuanDan | 0.463 | **0.598** | 0.15 | 0.112 | -0.390 | 0.637 | 0.88 | 0.96 |
| Riichi | 0.142 | 0.000 | **1.42** | 0.060 | -0.242 | 0.757 | 0.95 | -1.07 |
| Uno | 0.234 | 0.000 | 0.04 | **0.160** | -0.059 | 0.637 | 1.14 | -0.47 |
| Gin Rummy | 0.073 | 0.000 | 0.06 | 0.112 | **0.208** | 0.637 | -0.19 | 3.08 |
| Leduc | 0.172 | 0.000 | 0.10 | 0.052 | -0.515 | **1.244** | 2.47 | **6.98** |
| Limit | 0.167 | 0.000 | 0.13 | 0.052 | -0.469 | 1.105 | **2.84** | 6.86 |
| Nolimit | 0.170 | 0.000 | 0.04 | 0.056 | -0.198 | 1.000 | 2.06 | 4.92 |
| Mix | 0.870 | 0.661 | 1.38 | 0.164 | 0.186 | 1.244 | 2.77 | 6.02 |

Table 5: Influence between different games using Llama model.

| Model/Game | DouDizhu | GuanDan | Riichi | Uno | Gin Rummy | Leduc | Limit | Nolimit |
|---|---|---|---|---|---|---|---|---|
| DouDizhu | **0.723** | 0.010 | 0.10 | 0.060 | -0.460 | 0.637 | 1.14 | -1.21 |
| GuanDan | 0.447 | **0.629** | 0.02 | 0.136 | -0.362 | -0.068 | -0.14 | 2.34 |
| Riichi | 0.063 | 0.000 | **1.33** | 0.052 | -0.298 | 0.282 | 0.50 | -5.15 |
| Uno | 0.111 | 0.000 | 0.06 | **0.176** | -0.302 | 0.637 | 1.14 | 2.20 |
| Gin Rummy | 0.075 | 0.000 | 0.06 | 0.016 | **0.196** | 0.637 | 1.12 | 2.88 |
| Leduc | 0.142 | 0.000 | 0.08 | 0.048 | -0.416 | **1.244** | 2.41 | 6.02 |
| Limit | 0.125 | 0.000 | 0.04 | 0.096 | -0.411 | 1.232 | **3.02** | 5.05 |
| Nolimit | 0.114 | 0.000 | 0.06 | 0.004 | -0.499 | 0.648 | 1.53 | **6.24** |
| Mix | 0.882 | 0.698 | 1.31 | 0.252 | 0.191 | 1.244 | 2.89 | 5.77 |

Table 6: Influence between different games using GLM model.

**Influence Between Different Games.** To explore the mutual influence between different games, we evaluate the model fine-tuned on one game across the other seven games. The results are shown in Table 4,5,6. Compared to models trained on other games, the model trained on GuanDan also performs well on DouDizhu. This indicates that GuanDan has a positive influence on DouDizhu. Additionally, we can see that there are also positive influences among the three games, Leduc Hold'em, Limit Texas Hold'em, and No-limit Texas Hold'em. We claim that if the rules of two games are more similar, there tends to be greater knowledge transfer between them, for example, DouDizhu and

GuanDan, which have similar rules. Similarly, the three poker games, Leduc Hold'em, Limit Texas Hold'em, and No-limit Texas Hold'em, exhibit more knowledge transfer due to their similar game rules. Compared to DouDizhu and GuanDan, the rules of these three games are more similar, leading to a more significant transfer effect. Therefore, **game rules primarily dictate knowledge transfer between different games.**

We also compare the models fine-tuned on a single game with those fine-tuned on all games. The comparison results for Doudizhu and Guandan are shown in Figure 5. Because the card-playing rules of Doudizhu and Guandan are very similar, the performance of the mixed fine-tuned models improves further on both games compared to the models fine-tuned on each game individually. This indicates that Doudizhu and Guandan can mutually enhance each other's performance. However, we also observe that the performance of the mixed fine-tuned models declined on the other six games compared to the individually fine-tuned models. This suggests that there is a conflict between Doudizhu and Guandan and the other six games.

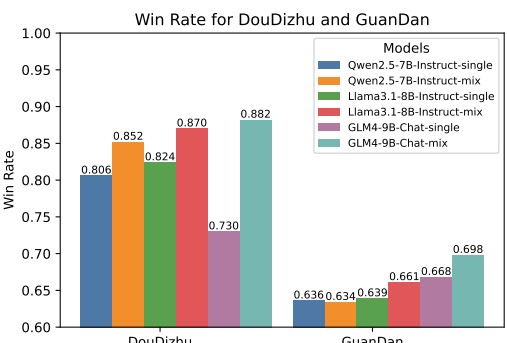

Figure 5: Comparison between models fine-tuned on a single game and models fine-tuned on all games.

**Different Model Sizes.** We train mixed models of different sizes on Qwen2.5. Figure 6 shows the performance of these models on Doudizhu and Guandan. The performance improves as the number of model parameters increases.

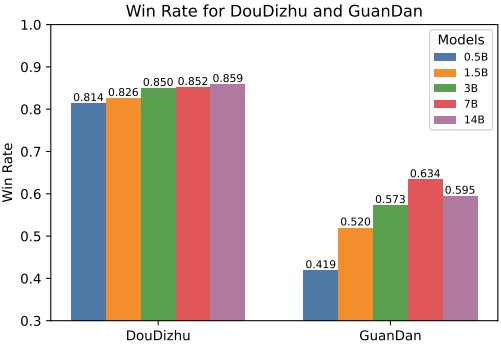

### 4.4 RQ3. *Can LLMs maintain their general capabilities while mastering complex games?*

**Experimental Design.** To test whether the models lose their general capabilities after mastering the games, we evaluate the models' performance of general knowledge question answering, mathematics, and coding before and after fine-tuning, using MMLU-Pro [12], Math-500 [27], and HumanEval [28] benchmarks. If general capabilities

Figure 6: Performance of models with different sizes on DouDizhu and GuanDan.

decline after fine-tuning on games, can further fine-tuning on knowledge, mathematics, and coding data help restore these general capabilities to some extent?

**Results and Analysis of General Benchmarks.**
The results are shown in Figure 7. The mixed models fine-tuned on all games show significant declines in their abilities in knowledge-based question answering, mathematics, and coding. We then further fine-tune the game model on a mixed dataset composed of knowledge data, mathematics data, coding data, and game data. The proportions of these four types of data were 20k, 20k, 20k, and 8k, respectively. The 8k game data consists of 8 games, with 1k data points for each game. The quantities and proportions are chosen by referring to previous work on general knowledge recovery [45]. The knowledge data, mathematics data, and coding data are taken from part of Tulu-3's post-training data [46], as this model has made

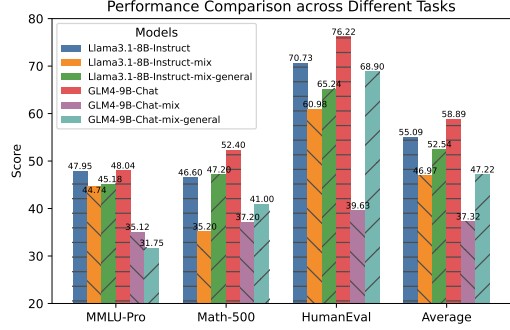

Figure 7: Evaluation results of different models on general benchmarks.

all its post-training data open source. The evaluation results of the model fine-tuned with the general data are shown in Figure 7. As shown in the table, by further fine-tuning on specific types of data,

the model can restore its ability in specific areas to some extent, as demonstrated in the paper with knowledge-based question answering, mathematics, and coding capabilities.

| Model | DouDizhu | GuanDan | Riichi | Uno | Gin Rummy | Leduc | Limit | Nolimit |
|---|---|---|---|---|---|---|---|---|
| Llama3.1-8B-Instruct-mix | 0.870 | 0.661 | 1.38 | 0.164 | 0.186 | 1.24 | 2.77 | 6.02 |
| Llama3.1-8B-Instruct-mix-general | 0.864 | 0.647 | 1.08 | 0.208 | 0.208 | 1.24 | 2.77 | 6.91 |
| GLM4-9B-Chat-mix | 0.882 | 0.698 | 1.31 | 0.252 | 0.191 | 1.24 | 2.89 | 5.77 |
| GLM4-9B-Chat-mix-general | 0.874 | 0.645 | 1.38 | 0.152 | 0.205 | 1.24 | 2.89 | 6.65 |

Table 7: Performance of fine-tuned models on games. Mix refers to models fine-tuned on the mixed training set composed of data from all games. General refers to models fine-tuned on the mixed training set composed of the knowledge data, mathematics data, and coding data.

**Results and Analysis of Games.** To evaluate the impact of general data fine-tuning on game performance, we provid the performance of the models on all games before and after fine-tuning with general data in Table 7. From the table, it can be seen that the model's performance on games has remained mostly unchanged (slight improvements or stability in 5 games, and slight decreases in 3 games), indicating that the model regains a certain level of general capability while maintaining its gaming ability.

**Different Model Types.** In the three non-gaming benchmarks (MMLU-Pro, Math-500, and HumanEval), after fine-tuning on game data, GLM exhibites greater performance degradation compared to LLaMA. Furthermore, after fine-tuning on general-purpose data, GLM showes a lower degree of recovery on all three benchmarks relative to LLaMA, particularly on MMLU-Pro. This indicates that LLaMA is better than GLM at maintaining general capabilities, especially in retaining general knowledge. This may be related to the differences in training data and training methods used by the two models.

## 4.5 Discussion on the advantages of LLMs compared to specialized game AI

**Comparison of computation and data.** We want to compare the amount of computation and data required for fine-tuning versus training a game AI system from scratch. However, due to the insufficient information disclosed about these game AI systems and differences in hardware environments, conducting a fair comparison is infeasible. Nevertheless, we have tried to list some comparative information in Appendix A.1. This information does not directly demonstrate the advantages of LLMs in terms of computation and data requirements during training.

However, the key aspect we aim to highlight is that the greatest advantage of LLMs lies in their nature as general-purpose learners. To achieve strong performance in games, both traditional reinforcement learning approaches and LLMs require the selection of appropriate game features. In this regard, both approaches are similar. Nevertheless, traditional reinforcement learning methods require the design of network architectures that are tailored to the chosen features. Different games employ different features, meaning that each game necessitates a specially designed network architecture—a step that is often the most labor-intensive. LLMs, on the other hand, eliminate the need for this step of network structure design and can be applied across all games without modification. We only need to design templates for each game. This flexibility is the foremost advantage of LLMs. For example, DouZero, DanZero, and Mortal have network architectures specifically designed for individual games. DouZero even requires separate designs for the two roles in the game. In contrast, LLMs can perform well in three different games using the same architecture. Thus, we argue that the general learning ability of LLMs represents their most significant advantage.

## 5 Conclusion

In this paper, we explore the potential of large language models (LLMs) to master complex card games, evaluating their performance through fine-tuning on carefully selected high-quality gameplay interaction data. We explore three key research questions concerning LLMs' ability to master complex card games, their capacity to learn multiple games simultaneously, and the impact of game mastery on their general capabilities. Our study reveals that LLMs have the potential to achieve strong performance in complex card games, while also handling multiple games at once and retaining significant portions of their general capabilities.

## Acknowledgements

This research is supported by the National Natural Science Foundation of China (Grant No. 62406163).

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

# A Appendix

## A.1 Comparison of computation and data

Among the three strong game AI models, Mortal, a Mahjong AI, does not have corresponding published papers, nor does its Git repository specify the computation and data required for training. Both DouZero and DanZero have published papers. Below is a comparison of our fine-tuned models with these two models in terms of hardware environment, training time, and data volume:

**DouZero:** According to their paper, DouZero was trained on a single server with 2 Intel(R) Xeon(R) Silver 4214R CPUs and 4 1080 Ti GPUs for 30 days; the data volume was not specified.

**LLM-Dou:** For the model in Figure 2a of our paper, we fine-tuned using a single server equipped with 2 Intel(R) Xeon(R) Platinum 8476C CPUs and 8 H800 GPUs on a dataset with 1 million samples. The fine-tuning times for the three models are as follows: Qwen2.5-7B: 11 hours; Llama3.1-8B: 12 hours; GLM4-9B: 14 hours.

**DanZero:** According to their paper, DanZero was trained on a server with 4 Intel(R) Xeon(R) Gold 6252 CPUs and a GeForce RTX 3070 GPU for 30 days; the data volume was not specified.

**LLM-Dan:** For the model in Figure 2b of our paper, we fine-tuned using a single server equipped with 2 Intel(R) Xeon(R) Platinum 8476C CPUs and 8 H800 GPUs on a dataset with 1 million samples. The fine-tuning times for the three models are as follows: Qwen2.5-7B: 21 hours; Llama3.1-8B: 25 hours; GLM4-9B: 29 hours.

## A.2 Evaluation on more general benchmarks

We provide the results of the models on four other common benchmarks (GQPA-Diamond, AIME2024, LiveCodeBench, IFEval) before and after fine-tuning on general mixed data in Table 8. From the table, it can be seen that if the general mixed data does not include a specific type of data, the model's corresponding capability will not be restored (after fine-tuning on the general mixed data, the performance of both models declined on the instruction-following benchmark).

| Model | MMLU-Pro | Math-500 | HumanEval | GQPA-Diamond | AIME2024 | LiveCodeBench | IFEval | Average First_Three | Average ALL |
|---|---|---|---|---|---|---|---|---|---|
| Llama-3.1-8B-Instruct | 47.95 | 46.60 | 70.73 | 21.21 | 6.67 | 20.25 | 74.68 | 55.09 | 41.16 |
| Llama-3.1-8B-Instruct-mix | 44.74 | 35.20 | 60.98 | 26.77 | 6.67 | 17.75 | 74.31 | 46.97 | 38.06 |
| Llama-3.1-8B-Instruct-mix-general | 45.18 | 47.20 | 65.24 | 27.27 | 10.00 | 13.50 | 68.95 | 52.54 | 39.62 |
| GLM-4-9B-Chat | 48.04 | 52.40 | 76.22 | 26.26 | 0.00 | 18.00 | 69.13 | 58.89 | 41.44 |
| GLM-4-9B-Chat-mix | 35.12 | 37.20 | 39.63 | 26.26 | 0.00 | 13.75 | 63.40 | 37.32 | 30.77 |
| GLM-4-9B-Chat-mix-general | 31.75 | 41.00 | 68.90 | 20.20 | 0.00 | 16.25 | 56.01 | 47.22 | 33.44 |

Table 8: Evaluation results of different models on more general benchmarks.

## A.3 Prompt Template

**Figure 1: Prompt Template of DouDizhu**

```
You are now a player in a game of Dou Dizhu (Fight the Landlord). The game rules
↪   are as follows:

1. The game is played by three players with a standard 54-card deck including a
↪   red joker and a black joker.
2. There are three roles in the game: landlord, landlord_down (farmer down of
↪   landlord), and landlord_up (farmer up of landlord).
3. After bidding, one player becomes the ''landlord'' who receives an extra three
↪   cards. The other two players are the ''peasants'' who work together to defeat
↪   the landlord.
```

4. In each round, the starting player must play a card or a valid combination of
↪   cards.
5. The other two players can choose to either follow with a higher-ranked card or
↪   combination, or pass.
6. If two consecutive players pass, the round ends and the player with the
↪   highest rank in that round starts the next round.
7. The objective is to be the first player to get rid of all the cards in hand.

The cards and comparison are as follows:
1. Individual cards are ranked. Colored Joker > Black & White Joker > 2 > Ace (A)
↪   > King (K) > Queen (Q) > Jack (J) > 10 > 9 > 8 > 7 > 6 > 5 > 4 > 3.
2. The Rocket (Red Joker and Black Joker) and the Bomb are groups of cards that
↪   work differently in terms of game play.
3. Compare only the same Category. Compare only the Chains with the same length.
↪   Compare the rank in the Primal cards only. Jokers and 2 are non-consecutive
↪   cards.
4. The type of card combination: Solo, Solo Chain (5), Pair, Pair Chain (3),
↪   Trio, Trio Chain (2), Trio with Solo, Trio Chain with Solo, Trio with Pair,
↪   Trio Chain with Pair, Bomb, Four with Dual solo, Four with Dual pair.

Your task is to make the best decision in each playing round. I will provide you
↪   with the following information:

Turn number:
%s

1. Your role:
%s

2. Your current hand cards:
%s

3. The union of the hand cards of the other two players:
%s

4. The most recent valid move:
%s

5. The played cards so far:
%s

6. The number of cards left for each player:
%s

7. The number of bombs played so far:
%s

8. The historical moves:
%s

```
9. The legal actions for the current move:
%s

Please tell me what cards you want to play in JSON format based on the provided
↪    information. The JSON should contain an "action" key with a value chose from
↪    legal actions.
If you choose to play cards, the value should contain the array of cards you want
↪    to play; if you choose to pass, the value should be empty array.

Output format examples:
Playing cards: {"action": [3, 3, 3]}
Passing: {"action": []}

Please provide the corresponding JSON action based on the given information.
```

**Figure 2: Prompt Template of GuanDan**

```
You are now a player in a game of Guandan. The game rules are as follows:

1. The game is played by four players in partnerships, sitting opposite each
↪    other.
2. The deck consists of two standard international decks with Jokers, totaling
↪    108 cards.
3. The objective is to play higher combinations of cards to empty your hand
↪    before your opponents.
4. If your team completes the game first, you will advance in levels; the
↪    ultimate goal is to win on Level A.
5. Card ranks in increasing order are: 2, 3, 4, 5, 6, 7, 8, 9, 10, J, Q, K, A.
6. There are four suits (Spades, Hearts, Diamonds, Clubs) and four Jokers (two
↪    red, two black).
7. Players take turns in counterclockwise order, starting from a player who plays
↪    any combination of cards.
8. Other players must play higher cards of the same type or a higher combination,
↪    or they must pass.
9. The game continues until three players have finished their cards.
10. Players are given titles based on the order they finish: Banker, Follower,
↪    Third, and Dweller.

The special cards and comparison are as follows:
1. Level Cards: The level number of the leading team determines the level cards.
↪    The level cards rank above aces but below jokers. For example, if the leading
↪    team is at level 6, then sixes are the level cards and rank above A.
2. Wild Cards: The two level cards in hearts are wild. During the round, they can
↪    be played as any card, except jokers, to form a combination with other cards.
↪    However, they only count as normal, non-wild cards when played as a single
↪    card. For example, when the level in the round is 7, the 7 of hearts can make
↪    a 4-bomb when combined with three 8s.
```

3. Normal Comparison: The normal comparison of the cards is from high to low in
↪  the order of red joker, black joker, A, K, Q, J, 10, 9, 8, 7, 6, 5, 4, 3, 2.
↪  It applies when comparing with a single card, pair, triple, tube, plate,
↪  straight, bomb, and straight flush. Specially, full house compares the triple
↪  in the combination only.
4. Bomb Comparison: Bomb depends on its number of cards. The smallest is a 4-bomb
↪  of 2s and the largest is an 8-bomb of aces. However, a 5-bomb of 2s is larger
↪  than a 4-bomb of aces. A bomb ranks above: single card, pair, triple, tube,
↪  plate, full house, straight. A straight flush is regarded as a bomb that
↪  ranks above a 4 or 5-card bomb, except the joker bomb. A bomb with 6 or more
↪  cards ranks above a straight flush. Straight flushes rank according to their
↪  largest card regardless of suits. The joker bomb is the largest bomb in the
↪  game.

The representation of cards and card types is as follows:
1. **Cards**: Represented by a two-character string, such as 'S2' which means
↪  Spade 2. Detailed description below:
    - **Suits**: Spades, Hearts, Clubs, and Diamonds are represented by the
    ↪  characters S, H, C, and D respectively. Specifically, the suit for the
    ↪  small Joker is S, and for the big Joker, it is H.
    - **Ranks**: A, 2, 3, 4, 5, 6, 7, 8, 9, 10, J, Q, K are represented by A, 2,
    ↪  3, 4, 5, 6, 7, 8, 9, T, J, Q, K respectively. That is, the rank 10 is
    ↪  represented by the character T. Specifically, the rank for the small Joker
    ↪  is represented by the character B, and for the big Joker, it is
    ↪  represented by the character R.
    For example, 'S2' represents Spade 2, 'HQ' represents Heart Q; 'SB' represents
    ↪  the small Joker, 'HR' represents the big Joker, 'PASS' indicates a pass.

2. **Card Types**: [Type, Rank, Cards]
    A card type is represented by a list of three fixed parts: Type, Rank, and
    ↪  Cards.
    - **Type**: The type of card combination, represented as a string with
    ↪  possible values of ['Single', 'Pair', 'Trips', 'ThreePair',
    ↪  'ThreeWithTwo', 'TripsPair', 'Straight', 'Boom', 'PASS', 'tribute',
    ↪  'back'].
    - **Rank**: The rank of the highest card or representative rank in the
    ↪  combination, with possible values of ['A', '2', '3', '4', '5', '6', '7',
    ↪  '8', '9', 'T', 'J', 'Q', 'K', 'B', 'R', 'PASS'].
    - **Cards**: The actual cards involved in the combination, represented as a
    ↪  list.
    Examples:
    - A single Diamond 5 is represented as: ['Single', '5', ['D5']].
    - A pair of 4s is represented as: ['Pair', '4', ['H4', 'C4']].
    - PASS: ['PASS', 'PASS', 'PASS'].

Your task is to make the best decision in each playing round. I will provide you
↪  with the following information:

1. Your position:

```
%s

2. Your current hand:
%s

3. Remaining cards of other players:
%s

4. Last action of other players:
%s

5. Last action of the teammate:
%s

6. Number of cards left for other players:
%s

7. Cards played by the down player:
%s

8. Cards played by the teammate:
%s

9. Cards played by the up player:
%s

10. Self rank:
%s

11. Opponent rank:
%s

12. Current rank:
%s

13. Legal actions:
%s

Please tell me your action in JSON format based on the provided information. The
↪  JSON should contain an "action" key with a value chose from legal actions.

Output format examples:
Playing a card: {"action": ["Single", "9", ["H9"]]}

Please provide the corresponding JSON action based on the given information.
```

**Figure 3: Prompt Template of Riichi Mahjong**

```
You are now a player in a game of Riichi Mahjong. The game rules are as follows:

1. The game uses 136 tiles divided into three suits (Pin, Sou, Wan) and honor
↪   tiles, which include wind and dragon tiles.
2. The tiles are mixed and arranged into four walls, two tiles high and 17 tiles
↪   wide.
3. Players draw and discard tiles to form valid groups (mentsu) of triplets
↪   (Pon), sequences (Chii), or quads (Kan).
4. A hand can be completed to declare a win by forming four groups and a pair.
5. Special rules include Riichi (declaring ready with a closed hand) and Dora
↪   indicators (bonus tiles).
6. Players can call tiles discarded by others to make open sets, making their
↪   hands open or closed.

All possible actions are: 'dahai: x', 'reach', 'chi_low', 'chi_mid', 'chi_high',
↪   'pon', 'kan', 'hora', 'ryukyoku', 'pass'.
'dahai: x': discard tile x.
'reach': declare a ready hand (riichi).
'chi_low', 'chi_mid', 'chi_high': Create a meld by completing a sequence, using
↪   the discarded tile.
'pon': create a three-of-a-kind meld using the discarded tile.
'kan': create a four-of-a-kind meld. This can be done in several ways: by adding
↪   a tile to an existing three-of-a-kind meld, using a discarded tile to
↪   complete a four-of-a-kind, or declaring a concealed four-of-a-kind by having
↪   four identical tiles in hand.
'hora': declare a win.
'ryukyoku': declare an aborted game or a draw.
'pass': Opt not to take any action or declaration. This can mean passing on a
↪   chance to chi, pon, kan, or win (hora).

Your task is to make the best decision in each playing round. I will provide you
↪   with the following information:

1. Your identifier:
%s

2. bakaze:
%s

3. jikaze:
%s

4. kyoku:
%s

5. honba:
```

```
%s

6. kyotaku:
%s

7. oya:
%s

8. Scores:
%s

9. Your rank:
%s

10. at turn:
%s

11. title left:
%s

12. shanten:
%s

13. my hands:
%s

14. wait tiles:
%s

15. dora indicators:
%s

16. dora owned:
%s

17. akas in your hand:
%s

18. doras seen:
%s

19. akas seen:
%s

20. tiles seen:
%s

21. ankan candidates:
%s
```

```
22. kakan candidates:
%s

23. kawa overview:
%s

24. fuuro overview:
%s

25. ankan overview:
%s

26. last tedashis:
%s

27. riichi sutehais:
%s

28. last self tsumo:
%s

29. last kawa tile:
%s

30. riichi declared:
%s

31. riichi accepted:
%s

32. can riichi:
%s

33. is riichi:
%s

34. at furiten:
%s

35. is menzen:
%s

36. Legal actions:
%s
```

Please tell me your action in JSON format based on the provided information. The
↪  JSON should contain an "action" key with a value chose from legal actions.

```
Output format examples:
Playing a card: {"action": "dahai: x"}

Please provide the corresponding JSON action based on the given information.
```

**Figure 4: Prompt Template of Uno**

```
You are now a player in a game of UNO. The game rules are as follows:

1. The game is played with a specially designed deck.
2. There are 2 players in the game.
3. Each player starts with seven cards dealt face down.
4. The top card from the Draw Pile is placed in the Discard Pile to start the
↪   game.
5. Players take turns matching the card in the Discard Pile by number, color, or
↪   symbol/action.
6. If a player has no matching card, they must draw a card from the Draw Pile.
7. If the drawn card can be played, the player must play it; otherwise, they keep
↪   the card.
8. The objective is to be the first player to get rid of all the cards in hand.

The deck of UNO includes 108 cards:
25 in each of four color suits (red, yellow, green, blue), each suit consisting
↪   of one zero, two each of 1 through 9, and two each of the action cards
↪   "Skip", "Draw Two", and "Reverse".
The deck also contains four "Wild" cards and four "Wild Draw Four".

Action or Wild cards have the following effects:
- Skip: Next player in the sequence misses a turn.
- Draw Two: Next player in the sequence draws two cards and misses a turn.
- Reverse: Order of play switches directions.
- Wild: Player declares the next color to be matched.
- Wild Draw Four: Player declares the next color to be matched; next player in
↪   sequence draws four cards and misses a turn.

Your task is to make the best decision on your turn. I will provide you with the
↪   following information:

Current step:
%s

1. Your position:
%s

2. Your hand:
%s

3. The top card in the Discard Pile:
```

```
%s

4. Played_cards:
%s

5. Number of cards left for each player:
%s

6. History actions of all players:
%s

7. Legal actions:
%s

Please tell me your action in JSON format based on the provided information. The
↪   JSON should contain an "action" key with a value chose from legal actions.
The value should be an card or "draw" if you want to draw a card.

Output format examples:
Drawing a card: {"action": "draw"}

Please provide the corresponding JSON action based on the given information.
```

---

**Figure 5: Prompt Template of Gin Rummy**

```
You are now a player in a game of Gin Rummy. The game rules are as follows:

1. The game is played by two players using a standard 52-card deck (ace is low).
2. The dealer deals 11 cards to the opponent and 10 cards to himself.
3. The non-dealer discards first. During each turn, you can pick up the discard
↪   or draw from the face-down stock, then discard a card.
4. Players try to form melds of 3 or more cards of the same rank or 3 or more
↪   cards of the same suit in sequence.
5. If the deadwood count (the value of non-melded cards) is 10 or less, a player
↪   can knock. If all cards can be melded, the player can gin.
6. If a player knocks or gins, the hand ends, and scores are determined. The
↪   opponent can lay off deadwood cards to extend melds of the knocker.
7. The score is the difference between the deadwood counts. If the score is
↪   positive, the knocker receives it; if zero or negative, the opponent receives
↪   the score plus a 25-point undercut bonus.
8. If neither player knocks or gins, they continue drawing and discarding cards.
↪   If the stockpile is reduced to two cards, the hand is declared dead.

All possible actions are: "draw_card", "pick_up_discard", "gin", "discard x",
↪   "knock x", "declare_dead",  "score N", or "score S".
"draw_card": Draw a card from the stockpile.
"pick_up_discard": Pick up the top card from the discard pile.
"gin": Declare gin.
```

```
"discard x": Discard a card from your hand.
"knock x": Knock a card from your hand.
"declare_dead": Declare dead.
"score N": Score player 0.
"score S": Score player 1.

Your task is to make the best decision in each phase of the game. I will provide
↪  you with the following information:

Current step:
%s

1. Your id:
%s

2. Your hand cards:
%s

3. Top card in the discard pile:
%s

4. Other cards in the discard pile:
%s

5. Opponent known cards:
%s

6. Left card number of stock pile:
%s

7. History actions of all players:
%s

8. Legal actions:
%s

Please tell me your action in JSON format based on the provided information. The
↪  JSON should contain an "action" key with a value chose from legal actions.

Output format examples:
Discarding a card: {"action": "discard 3S"}

Please provide the corresponding JSON action based on the given information.
```

**Figure 6: Prompt Template of Leduc Hold'em**

```
You are now a player in a game of Leduc Hold'em. The game rules are as follows:
```

1. The deck consists of only two pairs of King, Queen and Jack (6 cards in
↪ total).
2. There are two players in the game.
3. The game has two rounds with a two-bet maximum.
4. Raise amounts are 2 in the first round and 4 in the second round.
5. In the first round, each player puts 1 unit in the pot and is dealt one card.
6. In the second round, one public card is revealed.
7. The winner is determined by matching the player's card with the public card or
↪ having the highest rank.

All possible actions are: "fold", "call", "raise", or "check".

Your task is to make the best decision in each betting round. I will provide you
↪ with the following information:

Round number:
%s

1. Your position:
%s

2. Your hand:
%s

3. Public card  (if in round 2):
%s

4. Your chips in the pot:
%s

5. All chips in the pot:
%s

6. Number of raises so far in two rounds:
%s

7. History actions of all players:
%s

8. Legal actions:
%s

Please tell me your action in JSON format based on the provided information. The
↪ JSON should contain an "action" key with a value chose from legal actions.

Output format examples:
Folding: {"action": "fold"}
Calling: {"action": "call"}
Raising: {"action": "raise"}

```
Checking: {"action": "check"}

Please provide the corresponding JSON action based on the given information.
```

**Figure 7: Prompt Template of Limit Texas Hold'em**

```
You are now a player in a game of Limit Texas Hold'em. The game rules are as
↪  follows:

1. The deck consists of 52 cards.
2. There are multiple players in the game.
3. Each player is dealt two face-down cards (hole cards).
4. There are five community cards dealt in three stages: the flop (3 cards), the
↪  turn (1 card), and the river (1 card).
5. There are four betting rounds: pre-flop, flop, turn, and river.
6. In each round, players can choose to "call", "check", "raise", or "fold".
7. This is a fixed limit game, so raises are of a fixed amount.
8. The number of raises in each round is limited to 4.
9. The winner is determined by the best five-card hand using any combination of
↪  hole cards and community cards.

Texas Hold'em hands are ranked from highest to lowest as follows:
Royal Flush: A, K, Q, J, 10 all of the same suit.
Straight Flush: Five consecutive cards of the same suit. Higher top card wins.
Four of a Kind: Four cards of the same rank. Higher rank wins; if same, compare
↪  fifth card.
Full House: Three cards of one rank and two cards of another rank. Higher
↪  three-card rank wins; if same, compare the two-card rank.
Flush: Five non-consecutive cards of the same suit. Compare the highest card,
↪  then the second-highest, and so on.
Straight: Five consecutive cards of different suits. Higher top card wins.
Three of a Kind: Three cards of the same rank. Higher rank wins.
Two Pair: Two cards of one rank and two cards of another rank. Compare the higher
↪  pair first, then the lower pair, and then the fifth card.
One Pair: Two cards of the same rank. Compare the pair first, then the highest
↪  non-paired card, then the second highest, and so on.
High Card: If no hand can be formed, the highest card wins. If the highest cards
↪  are the same, compare the second highest, and so on.
If the hands are of equal rank, the pot is split.

All possible actions are: "fold", "call", "raise", or "check".

Your task is to make the best decision in each betting round. I will provide you
↪  with the following information:

Current betting round:
%s
```

```
1. Your position:
%s

2. Your hole cards:
%s

3. Community cards:
%s

4. Your chips in the pot:
%s

5. All chips in the pot:
%s

6. Number of raises so far in four rounds:
%s

7. History actions of all players:
%s

8. Legal actions:
%s

Please tell me your action in JSON format based on the provided information. The
↪   JSON should contain an "action" key with a value chose from legal actions.

Output format examples:
Folding: {"action": "fold"}
Calling: {"action": "call"}
Raising: {"action": "raise"}
Checking: {"action": "check"}

Please provide the corresponding JSON action based on the given information.
```

**Figure 8: Prompt Template of No-limit Texas Hold'em**

```
You are now a player in a game of No-limit Texas Hold'em. The game rules are as
↪   follows:

1. The deck consists of 52 cards.
2. There are multiple players in the game.
3. Each player is dealt two face-down cards (hole cards).
4. There are five community cards dealt in three stages: the flop (3 cards), the
↪   turn (1 card), and the river (1 card).
5. There are four betting rounds: pre-flop, flop, turn, and river.
6. In each round, players can choose to "call", "check", "raise", or "fold".
```

7. This is a no-limit game, so players can raise any amount from the minimum
↪ raise up to their entire stack.
8. The number of raises in each round is unlimited.
9. The winner is determined by the best five-card hand using any combination of
↪ hole cards and community cards.

Texas Hold'em hands are ranked from highest to lowest as follows:
Royal Flush: A, K, Q, J, 10 all of the same suit.
Straight Flush: Five consecutive cards of the same suit. Higher top card wins.
Four of a Kind: Four cards of the same rank. Higher rank wins; if same, compare
↪ fifth card.
Full House: Three cards of one rank and two cards of another rank. Higher
↪ three-card rank wins; if same, compare the two-card rank.
Flush: Five non-consecutive cards of the same suit. Compare the highest card,
↪ then the second-highest, and so on.
Straight: Five consecutive cards of different suits. Higher top card wins.
Three of a Kind: Three cards of the same rank. Higher rank wins.
Two Pair: Two cards of one rank and two cards of another rank. Compare the higher
↪ pair first, then the lower pair, and then the fifth card.
One Pair: Two cards of the same rank. Compare the pair first, then the highest
↪ non-paired card, then the second highest, and so on.
High Card: If no hand can be formed, the highest card wins. If the highest cards
↪ are the same, compare the second highest, and so on.
If the hands are of equal rank, the pot is split.

All possible actions are: "FOLD", "CHECK_CALL", "RAISE_HALF_POT", "RAISE_POT", or
↪ "ALL_IN".

Your task is to make the best decision in each betting round. I will provide you
↪ with the following information:

Current betting round:
%s

1. Your position:
%s

2. Your hole cards:
%s

3. Community cards:
%s

4. Your chips in the pot:
%s

5. All chips in the pot:
%s

```
6. Total chips of the pot:
%s

7. Remaining chips of all players:
%s

8. History actions of all players:
%s

9. Legal actions:
%s

Please tell me your action in JSON format based on the provided information. The
↪   JSON should contain an "action" key with a value chose from legal actions.

Output format examples:
Folding: {"action": "FOLD"}
Checking and calling: {"action": "CHECK_CALL"}
Raising half pot: {"action": "RAISE_HALF_POT"}
Raising pot: {"action": "RAISE_POT"}
Raising all remaining chips: {"action": "ALL_IN"}

Please provide the corresponding JSON action based on the given information.
```

## B   Limitations

Although the language models achieved performance close to that of strong game AIs, we found that
the inference time of LLMs in games is relatively longer compared to these AIs. This is because these
game AIs often have a smaller number of parameters, while most language models in our experiments
have parameter sizes in the billions (e.g., 7B). Although we used LoRA fine-tuning to reduce the
number of trainable parameters, inference still requires calculating all the parameters, resulting in
longer inference times.

## C   Broader Impact

This paper presents work whose goal is to advance the field of LLMs. Our work evaluates the learning
capabilities of large models through games and does not have negative societal impacts.

