# OpenReview forum: "Can Large Language Models Master Complex Card Games?"
_NeurIPS.cc/2025/Conference — NeurIPS 2025 poster_

### Official Review · Reviewer_2Ka4 · 2025-06-16

**Clarity:** 2
**Significance:** 3
**Originality:** 2
**Rating:** 4
**Confidence:** 4

**Summary:**

This article is an empirical study that extensively explores the learning capabilities of Large Language Models (LLMs) in complex card games. The author systematically evaluates 8 different complexity levels of card games and studies the potential of LLMs to master complex games through supervised fine-tuning on high-quality game data. The experimental results indicate that: (1) LLMs can approximate the performance of strong game AI through supervised fine-tuning; (2) Compared to previous RL-Based AI, LLMs can simultaneously master multiple complex card games; (3) Mastering complex games leads to a decline in general capabilities, but this can be mitigated by integrating general instruction data.

**Questions:**

1. Could the authors provide a direct comparison of win rates between LLM-based and RL-based strategies? Currently, the comparisons mainly involve rule-based models.

2. Have the authors considered evaluating the performance of reasoning-based models (such as O3, R1) on these card tasks?

3. Have you considered directly applying reinforcement learning to LLMs for card tasks? What are the potential advantages of this approach?

4. Regarding the performance decline of the 14B model, could you provide a more detailed analysis? For example, how does it perform on multi-task data? Can it better handle different tasks compared to the 7B model?

**Ethical Concerns:**

["NO or VERY MINOR ethics concerns only"]

**Final Justification:**

4 Borderline accept

**Limitations:**

See Strengths and Weaknesses.

**Quality:**

3

**Strengths And Weaknesses:**

**Strengths:**

1. Comprehensive experimental design: The author selected 8 card games of varying complexities, providing a comprehensive evaluation of LLMs' game learning capabilities.

2. Detailed ablation studies: The paper conducted various ablation experiments, including performance of commercial models, the impact of model size on performance in supervised fine-tuning, the influence of training data volume, and the interplay between tasks in multitask learning.

3. Practical value: Validates that LLMs with a parameter scale of 8 billion, after supervised fine-tuning with data generated by dedicated reinforcement learning AI, can achieve high performance in various card games.

4. Multi-dimensional analysis: Not only evaluated game performance but also analyzed the impact on general capabilities, as well as the knowledge transfer effects between different games.

5. Training cost comparison: The paper clearly compared the training resources and time costs between RL-Based and LLM-Based approaches.

**Weaknesses:**

1. Comparison benchmark limited: Primarily compared with rule-based models, lacking direct performance comparisons with original RL-based strategies.

2. Lack of reasoning model evaluation: Although understandable given time constraints at the project's initiation, there is a lack of evaluation of current mainstream reasoning models (such as O3, R1, etc.), which could be supplemented when time permits.

3. Unexplored potential of reinforcement learning: For card games with clear rules and rewards, the direct possibility of applying reinforcement learning to LLMs has not been explored. This could be an important research direction and may not require additional experiments.

4. Insufficient explanation for abnormal performance of the 14B model: The author attributes the performance decline of the 14B model to overfitting, but this explanation lacks persuasiveness and requires a more in-depth analysis.

5. Why can't general knowledge be trained together with the card game tasks? Will the sequence of training first on card games and then on general tasks reduce the performance of the card game tasks?

6. In the limitations section, there is no detailed explanation of the differences in reasoning time.

---

> ### Author Rebuttal · Authors · 2025-07-31
>
> ```
> W1: Primarily compared with rule-based models, lacking direct performance comparisons with original RL-based strategies.
> ```
> **Currently, the original RL-based strategies in multiple games are position-bound during training, meaning they can only be deployed in specific positions.** In five out of the eight games, reinforcement learning was used to generate data as teacher models (DouDizhu, GuanDan, Leduc Hold’em, Limit Texas Hold’em, No-limit Texas Hold’em). For GuanDan, the original RL-based strategies only provided pre-trained models for specific positions (a model trained for position 1 and position 3, with no models for position 2 and position 4; positions 1 and 3 form the first team, while positions 2 and 4 form the second team). In Leduc Hold’em, Limit Texas Hold’em, and No-limit Texas Hold’em, the models pre-trained using the RLCard default code were also trained for position 1. Therefore, the pre-trained models for these four games can only be used in position 1, which is why we did not make comparisons with the original RL-based strategies. For DouDizhu, DouZero[1] also trains different models for different positions and does not compare with its own strategy, but rather with other methods. We followed DouZero's comparison setup.
>
> [1] DouZero: Mastering DouDizhu with Self-Play Deep Reinforcement Learning
>
> ```
> W2: There is a lack of evaluation of current mainstream reasoning models (such as O3, R1, etc.), which could be supplemented when time permits.
> ```
> We have added the results for DeepSeek-R1 below (along with results for the other two models for comparison). From the table, it can be seen that DeepSeek-R1 shows improvements in most games compared to DeepSeek-V3. However, our mixed training model still outperforms DeepSeek-R1 in most games. The reasoning model takes significantly longer to evaluate compared to models without reasoning, and due to current time constraints, we have only supplemented the results for DeepSeek-R1.
> | Model            | DouDizhu | GuanDan | Riichi | Uno   | Gin Rummy | Leduc | Limit | Nolimit |
> |------------------|----------|---------|--------|-------|-----------|-------|-------|---------|
> | DeepSeek-V3      | 0.320    | 0.000   | 0.15   | 0.128 | 0.147     | 0.77  | 0.22  | 0.18    |
> | DeepSeek-R1      | 0.185    | 0.02    | 0.05   | 0.148 | 0.228     | 0.88  | 0.24  | 1.88    |
> | GLM4-9B-Chat-mix | 0.882    | 0.698   | 1.31   | 0.252 | 0.191     | 1.24  | 2.89  | 5.77    |
>
> ```
> W3: Unexplored potential of reinforcement learning. Have you considered directly applying reinforcement learning to LLMs for card tasks? What are the potential advantages of this approach?
> ```
> Currently, we have only evaluated the learning capability of LLMs through supervised instruction fine-tuning. In the future, we will explore whether reinforcement learning methods like GRPO can further enhance the learning ability of LLMs. The advantage of GRPO lies in its ability to automatically explore the chain of thought reasoning process between observations and card playing. Additionally, we are considering adopting an online reinforcement learning approach, where the first step involves using SFT to enable the model to grasp basic game strategies, and the second step involves using RL to allow the model to directly explore and learn in the environment online.
>
> ```
> W4: The author attributes the performance decline of the 14B model to overfitting, but this explanation lacks persuasiveness and requires a more in-depth analysis.
> ```
> We have added further analysis of the 14B model and found that it failed to balance the learning of the two roles, resulting in an average win rate lower than that of the 7B model. Below are the win rates of the 7B and 14B models when playing as the landlord and the farmer and average win rates. From the table, it can be seen that the **14B model performs better as the landlord (approaching the performance of the teacher model) but significantly worse as the farmer. This results in the average win rate of the 14B model being lower than that of the 7B model.** This is similar to why the Glm model performs worse on DouDizhu compared to Qwen and Llama (Line 240-244). The 14B model did not effectively balance the learning between the two roles and focused more on the landlord role, leading to weaker performance for the farmer role.
> | Model                | Landlord | Farmer | Avg. Win Rate |
> |----------------------|----------|--------|---------------|
> | Qwen2.5-7B-Instruct  | 0.828    | 0.784  | 0.806         |
> | Qwen2.5-14B-Instruct | 0.858    | 0.570  | 0.714         |
>
> ```
> W5: Why can't general knowledge be trained together with the card game tasks? Will the sequence of training first on card games and then on general tasks reduce the performance of the card game tasks?
> ```
> It is possible to train using both general data and game data together. Since we obtained a model trained on all data in our second experiment, we explored whether adding a small amount of general data to this model could restore its general capability. In the future, we will analyze the performance changes when training with both general data and game data together.
>
> ```
> W6: In the limitations section, there is no detailed explanation of the differences in reasoning time.
> ```
> Game AIs compared in our papers often have a smaller number of parameters, whereas most of the language models in our experiments have parameter sizes in the billions (e.g., 7B), which results in longer inference times.

---

> > ### Author Response · Authors · 2025-08-04
> > **Gentle Reminder from Authors**
> >
> > This is a gentle reminder that we are still awaiting your response regarding our rebuttal. We hope that our response has addressed your concerns and would like to hear from you. Please let us know if you have any further questions or feedback.

---

> > ### Comment · Reviewer_2Ka4 · 2025-08-06
> >
> > I would like to thank the author for the additional information. The experiments added addressed my concerns to some extent, and I think it is fair to improve my score slightly. However, the lack of evaluation of RL methods prevents me from directly considering this paper as acceptable.

---

### Official Review · Reviewer_Zt88 · 2025-06-24

**Clarity:** 4
**Significance:** 2
**Originality:** 3
**Rating:** 4
**Confidence:** 4

**Summary:**

This paper evaluates the learning ability of Large Language Models (LLMs) by fine-tuning them on a hight-quality dataset spanning 8 diverse card games, and compare their performances against the state-of-the-art agents like AlphaGo, AlphaZero, and MuZero. It also further examines how the fine-tuning impacts the models' general abilities and explore the strategies to restore the general capabilities of LLMs.

**Questions:**

1. The merged dataset spans from 50K to 950K samples per game. I wonder how sensitive is convergence to this imbalance? Have you tried balanced or curriculum sampling to ensure underrepresented games aren’t underfit?
2. You show that mixed-dataset tuning recovers general capabilities, can you provide per-game win rates post-mix to confirm that individual game skills weren’t degraded?
3. Why top reasoning LLMs are excluded? Including any of them could have clarified the true performance gap and dataset quality.

**Ethical Concerns:**

["NO or VERY MINOR ethics concerns only"]

**Final Justification:**

Supplementing the per-game win rates of the model after mixed-dataset tuning and including DeepSeek-R1 as baseline, act more as necessary justifications and supporting evidence that were previously missing, but is surely supports the claims of the paper. Thus, I would like to increase the clarity score as reward.

**Limitations:**

Yes.

**Paper Formatting Concerns:**

The paper is well-formatted.

**Quality:**

3

**Strengths And Weaknesses:**

#### **Strengths**
1. The authors utilizes strong game AIs to build a large, high-quality dataset that spans diverse scenarios.
2. The paper is well-written, analyses and discussions are comprehensive and easy to read.
3. Provides insightful experiments correlating fine-tuning data with in-game performance.
4. Explores how high-quality fine-tuning impacts both specialized gaming skills and broader general capabilities, and also the possibility of recovering the general capabilities of LLMs.
#### **Weaknesses**
1. Only open models are being fine-tuned, though closed models cannot be fine-tuned by LoRA, it can still be fine-tuned with the produced dataset to verify the quality of the data.
2. The paper omits comparisons with leading reasoning models, leaving the performance gap between the fine-tuned card-game models and top-tier baselines unquantified.
3. The highly uneven distribution of game samples in the merged 3.1 M point dataset (ranging from 50 K to 950 K per game) may prevent the model from converging on all tasks simultaneously.
4. After mixed-dataset fine-tuning restores some general capabilities, the paper fails to report per-game performance, so it remains unclear whether game-specific skills were preserved or sacrificed in the process.

---

> ### Author Rebuttal · Authors · 2025-07-31
>
> ```
> W1: Only open models are being fine-tuned, though closed models cannot be fine-tuned by LoRA, it can still be fine-tuned with the produced dataset to verify the quality of the data.
> ```
> The open-source models demonstrated strong performance after fine-tuning on the produced dataset, which has already validated the quality of the data. Additionally, the cost of fine-tuning with closed models is relatively high. Therefore, we did not use closed-source models for fine-tuning.
>
> ```
> W2: The paper omits comparisons with leading reasoning models, leaving the performance gap between the fine-tuned card-game models and top-tier baselines unquantified.
> ```
> We have added the results for DeepSeek-R1 below (along with results for the other two models for comparison). From the table, it can be seen that DeepSeek-R1 shows improvements in most games compared to DeepSeek-V3. However, our mixed training model still outperforms DeepSeek-R1 in most games. The reasoning model takes significantly longer to evaluate compared to models without reasoning, and due to current time constraints, we have only supplemented the results for DeepSeek-R1.
> | Model            | DouDizhu | GuanDan | Riichi | Uno   | Gin Rummy | Leduc | Limit | Nolimit |
> |------------------|----------|---------|--------|-------|-----------|-------|-------|---------|
> | DeepSeek-V3      | 0.320    | 0.000   | 0.15   | 0.128 | 0.147     | 0.77  | 0.22  | 0.18    |
> | DeepSeek-R1      | 0.185    | 0.02    | 0.05   | 0.148 | 0.228     | 0.88  | 0.24  | 1.88    |
> | GLM4-9B-Chat-mix | 0.882    | 0.698   | 1.31   | 0.252 | 0.191     | 1.24  | 2.89  | 5.77    |
>
> ```
> W3: The highly uneven distribution of game samples in the merged 3.1 M point dataset (ranging from 50 K to 950 K per game) may prevent the model from converging on all tasks simultaneously.
> ```
> We determined the data volume for each game based on the results from training on individual games. It is indeed possible that during mixed training, the model may not converge on all tasks simultaneously.
>
> ```
> W4: After mixed-dataset fine-tuning restores some general capabilities, the paper fails to report per-game performance, so it remains unclear whether game-specific skills were preserved or sacrificed in the process.
> ```
> Below, we have supplemented the performance of the model on games after training with general data (with results before training listed for comparison). From the table, it can be seen that the model's performance on games has remained mostly unchanged (slight improvements or stability in 5 games, and slight decreases in 3 games), indicating that the model has essentially regained a certain level of general capability while maintaining its gaming ability.
> | Model                             | DouDizhu | GuanDan | Riichi | Uno   | Gin Rummy | Leduc | Limit | Nolimit |
> |-----------------------------------|----------|---------|--------|-------|-----------|-------|-------|---------|
> | Qwen2.5-7B-Instruct-mix           | 0.852    | 0.634   | 1.08   | 0.108 | 0.177     | 1.24  | 2.66  | 4.86    |
> | Qwen2.5-7B-Instruct-mix-general   | 0.854    | 0.544   | 1.06   | 0.236 | 0.220     | 1.24  | 2.82  | 5.50    |
> | Llama3.1-8B-Instruct-mix          | 0.870    | 0.661   | 1.38   | 0.164 | 0.186     | 1.24  | 2.77  | 6.02    |
> | Llama-3.1-8B-Instruct-mix-general | 0.864    | 0.647   | 1.08   | 0.208 | 0.208     | 1.24  | 2.77  | 6.91    |
> | GLM4-9B-Chat-mix                  | 0.882    | 0.698   | 1.31   | 0.252 | 0.191     | 1.24  | 2.89  | 5.77    |
> | Glm-4-9B-Chat-mix-general         | 0.874    | 0.645   | 1.38   | 0.152 | 0.205     | 1.24  | 2.89  | 6.65    |
>
> ```
> Q1: The merged dataset spans from 50K to 950K samples per game. I wonder how sensitive is convergence to this imbalance? Have you tried balanced or curriculum sampling to ensure underrepresented games aren’t underfit?
> ```
> Due to the long training time and high costs associated with training on data from all games, we have currently only considered the most basic mixing strategy. Each game uses the data volume from when it converges during individual training to ensure sufficient learning for each game.

---

> > ### Comment · Reviewer_Zt88 · 2025-08-02
> >
> > Thanks for adding the DeepSeek-R1 baseline and efforts in addressing the weaknesses. However, my main concerns raised in Q2 and Q3 remain largely unaddressed, which I view as the more critical issues. While I also note that the determination of the convergence point may require further clarification, this is secondary. For now, I will keep my rating unchanged.

---

> > > ### Author Response · Authors · 2025-08-03
> > >
> > > Thank you very much for your feedback and valuable suggestions. Below, we will address the questions you raised. If you have any further questions, please feel free to ask us, and we will do our best to respond.
> > >
> > > ```
> > > Q2: You show that mixed-dataset tuning recovers general capabilities, can you provide per-game win rates post-mix to confirm that individual game skills weren’t degraded?
> > > ```
> > > We have supplemented per-game win rates of the model after mixed-dataset tuning.
> > >
> > > | Model                             | DouDizhu  | GuanDan | Riichi   | Uno       | Gin Rummy | Leduc    | Limit    | Nolimit  |
> > > |-----------------------------------|-----------|---------|----------|-----------|-----------|----------|----------|----------|
> > > | Qwen2.5-7B-Instruct-mix           | 0.852     | 0.634   | 1.08     | 0.108     | 0.177     | 1.24     | 2.66     | 4.86     |
> > > | Qwen2.5-7B-Instruct-mix-general   | **0.854** | 0.544   | 1.06     | **0.236** | **0.220** | **1.24** | **2.82** | **5.50** |
> > > | Llama3.1-8B-Instruct-mix          | 0.870     | 0.661   | 1.38     | 0.164     | 0.186     | 1.24     | 2.77     | 6.02     |
> > > | Llama-3.1-8B-Instruct-mix-general | 0.864     | 0.647   | 1.08     | **0.208** | **0.208** | **1.24** | **2.77** | **6.91** |
> > > | GLM4-9B-Chat-mix                  | 0.882     | 0.698   | 1.31     | 0.252     | 0.191     | 1.24     | 2.89     | 5.77     |
> > > | Glm-4-9B-Chat-mix-general         | 0.874     | 0.645   | **1.38** | 0.152     | **0.205** | **1.24** | **2.89** | **6.65** |
> > >
> > > From the table, it can be seen that the model's performance on games has remained mostly unchanged (slight improvements or stability in 5 games, and slight decreases in 3 games), indicating that **the model has essentially regained a certain level of general capability while most individual game skills weren’t degraded.**
> > >
> > > ```
> > > Q3: Why top reasoning LLMs are excluded? Including any of them could have clarified the true performance gap and dataset quality.
> > > ```
> > > We have added the results for DeepSeek-R1 below (along with results for the other two models for comparison). From the table, it can be seen that DeepSeek-R1 shows improvements in most games compared to DeepSeek-V3. However, **our mixed training model still outperforms DeepSeek-R1 in most games.**
> > >
> > > | Model            | DouDizhu | GuanDan | Riichi | Uno   | Gin Rummy | Leduc | Limit | Nolimit |
> > > |------------------|----------|---------|--------|-------|-----------|-------|-------|---------|
> > > | DeepSeek-V3      | 0.320    | 0.000   | 0.15   | 0.128 | 0.147     | 0.77  | 0.22  | 0.18    |
> > > | DeepSeek-R1      | 0.185    | 0.02    | 0.05   | 0.148 | 0.228     | 0.88  | 0.24  | 1.88    |
> > > | GLM4-9B-Chat-mix | 0.882    | 0.698   | 1.31   | 0.252 | 0.191     | 1.24  | 2.89  | 5.77    |
> > >
> > > When initially submitting the paper, we considered including evaluations of top reasoning LLMs. However, our preliminary experiments showed that reasoning LLMs have significantly longer inference times and are much more costly compared to non-reasoning LLMs. Due to these factors, we decided not to include evaluations of top reasoning LLMs. Currently, we have included the results of DeepSeek-R1, which have demonstrated the high quality of our dataset. We plan to add results for more reasoning LLMs when we have more time in the future.
> > >
> > > ```
> > > The merged dataset spans from 50K to 950K samples per game. The determination of the convergence point may require further clarification
> > > ```
> > > **We first trained the model separately on each game's data for a sufficient number of steps and determined when the model had converged based on win rate or rewards**, thereby identifying the data volume required for each game's convergence. Experimental results indicate that there are indeed significant differences in the data volume needed for convergence across different games. Based on the above experimental results, we roughly determine the amount of data required for each game to converge. We then sample data from the training datasets of each game according to this amount and merge them to obtain a mixed training set.
> > >
> > > We trained on the mixed training set for one epoch, and the experimental results showed that the number of steps needed to reach the highest win rate or rewards varied among the 8 games. DouDizhu and GuanDan had the highest win rates by the end of the training. This could be because these two games are the most difficult among eight games. **Since the goal of mixed training is to obtain a model that performs well across most games, we used the checkpoint at the end of one epoch as our final mixed model.**

---

> > > > ### Comment · Reviewer_Zt88 · 2025-08-04
> > > >
> > > > Thank you for supplementing the per-game win rates of the model after mixed-dataset tuning. However, I believe that including reasoning models should be standard practice and would serve well as baselines. As I mentioned in the strengths, I recognize the quality of the dataset—so these additional experiments act more as necessary justifications and supporting evidence that were previously missing. I hope both tables can be included in the final version to further improve the paper. While I will keep my overall rating unchanged, I am willing to raise the clarity score.

---

### Official Review · Reviewer_koKz · 2025-06-26

**Clarity:** 3
**Significance:** 3
**Originality:** 3
**Rating:** 5
**Confidence:** 5

**Summary:**

The paper evaluates LLMs across eight complex card games by finetuning them on high-quality gameplay data. The paper investigates three research questions: (i) can LLMs master complex card games? (ii) Can they master multiple games simultaneously? And (iii) do their general-purpose capabilities degrade if they are fine-tuned on gameplay data? The paper conducts an empirical investigation of these questions, and the results show that LLMs can master complex card games, but their general-purpose capabilities deteriorate as a result of finetuning.

**Questions:**

* How do you ensure that the training and evaluation data are different?
* Why do you attribute the poor performance of the 14B model in Figure 4 to overfitting and not some other problem? What does its performance on the training distribution look like?
* Why do all the finetuned models obtain a score of 1.24 for Leduc in Table 2?
* L121: Does a standard deck not have 52 cards?

**Ethical Concerns:**

["NO or VERY MINOR ethics concerns only"]

**Final Justification:**

The authors' rebuttal addressed most of my concerns, and the promised additional results and elaborations will greatly improve the paper. I have therefore increased my score.

**Limitations:**

Yes

**Paper Formatting Concerns:**

There are no major formatting issues in this paper.

**Quality:**

2

**Strengths And Weaknesses:**

**Strengths**

* The research questions are sensible and interesting and form a complete study.
* The experiments are comprehensive and executed well.
* The paper is well-written, easy to follow, and contains sufficient information to reproduce the results.

**Weaknesses**

* Some claims are misleading, incomplete, or incorrect
  * The paper claims that the base models “lack the strategic knowledge required for games, resulting in poor initial performance”. However, the paper provides no evidence that base models do not contain such “strategic knowledge”; it just shows that the models perform poorly when playing. Instead, the poor performance can be attributed to the “knowing-doing gap”, which was documented in other game environments (Paglieri et al., 2025; Ruoss et al., 2025) and states that LLMs perform poorly *even though* they contain a lot of (strategic) knowledge about these games.
  * The paper claims that LLMs “can master multiple complex card games simultaneously” and that training on multiple games can “mutually enhance each other’s performance”. However, the results in Table 3 actually show the opposite, since the models finetuned on a single game almost always perform better than the models trained on data from all games (except for DouDizhu), and sometimes quite substantially so, even though they were trained on roughly the same amount of data (“the amount of data required for each game to converge”). Consequently, if I have a FLOPS budget of X, I will obtain a better performance by training eight individual models, one for each game, with roughly X/8 FLOPS per model, than spending the entire budget on finetuning a single model across all games. Thus, since the individual models score quite a bit higher than the joint one, the LLMs cannot “master multiple games simultaneously”, and since there is no (or very little) transfer between games, it is better to train individual models than a general one.
* The experiment showing that some of the general-purpose capabilities can be recovered is ill-defined. Firstly, it is not very surprising that the performance of a fixed set of evals can be improved by finetuning on relevant data. However, the effects of that training process on other general-purpose capabilities are not shown. Secondly, the paper does not show how the model performs on the card games after some of the general-purpose capabilities have been “recovered”.
* The paper studies imitation learning on imperfect information games, which is susceptible to delusions (Ortega et al., 2021).
* The paper should discuss relevant related work on evaluating LLMs in game environments (Paglieri et al., 2025; Ruoss et al., 2025) as mentioned above.

I am very happy to increase my score if these weaknesses and the questions below are reasonably addressed.


**References**

Ortega et al. Shaking the foundations: delusions in sequence models for interaction and control. 2021.

Paglieri et al. BALROG: Benchmarking Agentic LLM and VLM Reasoning On Games. ICLR 2025.

Ruoss et al. LMAct: A Benchmark for In-Context Imitation Learning with Long Multimodal Demonstrations. ICML 2025.

---

> ### Author Rebuttal · Authors · 2025-07-31
>
> ```
> W1-1: The paper claims that the base models “lack the strategic knowledge required for games, resulting in poor initial performance”. However, the paper provides no evidence that base models do not contain such “strategic knowledge”; it just shows that the models perform poorly when playing. Instead, the poor performance can be attributed to the “knowing-doing gap”(Paglieri et al., 2025; Ruoss et al., 2025).
> ```
> **The focus of our work is to analyze whether models can achieve relatively good performance in complex games through learning.** We have not further analyzed the reasons for the poor performance of the base model. The referenced work you mentioned provides another possible reason. Therefore, we modify the statement to: "The poor initial performance is due to the base model's lack of strategic knowledge, or its inability to utilize existing strategic knowledge to make decisions in the environment."
>
> ```
> W1-2: The paper claims that LLMs “can master multiple complex card games simultaneously” and that training on multiple games can “mutually enhance each other’s performance”. However, the results in Table 3 actually show the opposite, since the models finetuned on a single game almost always perform better than the models trained on data from all games (except for DouDizhu).
> ```
> Our complete hypothesis is, **"LLMs can master multiple complex card games simultaneously, with performance augmentation for games with similar rules and conflicts for dissimilar ones"** (second finding in the abstract). We have also discussed this in detail in our experiments (Line 290-315). The point you mentioned, "that training on multiple games can mutually enhance each other’s performance," only applies to games with similar rules.
>
> Firstly, "LLMs can master multiple games simultaneously" refers to our model compared to API-based models (Table 2), where the model trained on all games surpasses the best API-based models in most games.
>
> Secondly, as analyzed in our paper (Line 300-315), DouDizhu and GuanDan are similar and can enhance each other, whereas these two games have significant rule differences compared to the other six games. In mixed training, the data quantity for DouDizhu and GuanDan is larger, which benefits from mixed training. Therefore, the performance of these two games after mixed training is better than training individually (Llama and Glm from Figure 5). This has already demonstrated that mixed training helps "LLMs can master multiple games simultaneously." We currently use only one heuristic data mixing ratio and believe that more refined adjustments in the future will allow the model to perform well across more games.
>
> ```
> W2: The experiment showing that some of the general-purpose capabilities can be recovered is ill-defined. Firstly,the effects of that training process on other general-purpose capabilities are not shown. Secondly, the paper does not show how the model performs on the card games after some of the general-purpose capabilities have been “recovered”.
> ```
> **General capability encompasses various aspects and currently lacks a precise definition.** Therefore, accurately measuring a model's general capability is relatively challenging. In our experiments, we investigated commonly used benchmarks from language model technical reports and selected typical benchmarks in three areas—knowledge-based Q&A, math, and code—as representatives to measure the model's general capability. In the future, we consider including other commonly used general capability evaluation benchmarks.
>
> Below, we have supplemented the performance of the model on games after training with general data (with results before training listed for comparison). From the table, it can be seen that the model's performance on games has remained mostly unchanged (slight improvements or stability in 5 games, and slight decreases in 3 games), indicating that **the model has essentially regained a certain level of general capability while maintaining its gaming ability.**
> | Model                             | DouDizhu | GuanDan | Riichi | Uno   | Gin Rummy | Leduc | Limit | Nolimit |
> |-----------------------------------|----------|---------|--------|-------|-----------|-------|-------|---------|
> | Qwen2.5-7B-Instruct-mix           | 0.852    | 0.634   | 1.08   | 0.108 | 0.177     | 1.24  | 2.66  | 4.86    |
> | Qwen2.5-7B-Instruct-mix-general   | 0.854    | 0.544   | 1.06   | 0.236 | 0.220     | 1.24  | 2.82  | 5.50    |
> | Llama3.1-8B-Instruct-mix          | 0.870    | 0.661   | 1.38   | 0.164 | 0.186     | 1.24  | 2.77  | 6.02    |
> | Llama-3.1-8B-Instruct-mix-general | 0.864    | 0.647   | 1.08   | 0.208 | 0.208     | 1.24  | 2.77  | 6.91    |
> | GLM4-9B-Chat-mix                  | 0.882    | 0.698   | 1.31   | 0.252 | 0.191     | 1.24  | 2.89  | 5.77    |
> | Glm-4-9B-Chat-mix-general         | 0.874    | 0.645   | 1.38   | 0.152 | 0.205     | 1.24  | 2.89  | 6.65    |
>
> ```
> W3: The paper studies imitation learning on imperfect information games, which is susceptible to delusions (Ortega et al., 2021). The paper should discuss relevant related work on evaluating LLMs in game environments (Paglieri et al., 2025; Ruoss et al., 2025) as mentioned above.
> ```
> Thank you for your suggestion. We will include discussions of these works in our related work section. As we mentioned in the introduction, a significant category of existing evaluations assesses model performance through prompting. The works by Paglieri et al., 2025, and Ruoss et al., 2025, fall into this category. Paglieri et al., 2025, evaluate both LLM and VLM in a gaming environment, whereas Ruoss et al., 2025, focus on assessing the impact of multimodal examples in context on performance. Ortega et al., 2021, emphasizes the LLM's ability to model causal relationships. Our work focuses on evaluating the learning ability of the model, hence there are substantial differences from these three works.
>
> ```
> Q1: How do you ensure that the training and evaluation data are different?
> ```
> Each game can control the initial hand data by specifying a random seed. We used different random seeds during training and evaluation to ensure that the hand data differed between these two phases.
>
> ```
> Q2: Why do you attribute the poor performance of the 14B model in Figure 4 to overfitting and not some other problem?
> ```
> We have added further analysis of the 14B model and found that it failed to balance the learning of the two roles, resulting in an average win rate lower than that of the 7B model. Below are the win rates of the 7B and 14B models when playing as the landlord and the farmer and average win rates. From the table, it can be seen that the **14B model performs better as the landlord (approaching the performance of the teacher model) but significantly worse as the farmer. This results in the average win rate of the 14B model being lower than that of the 7B model.** This is similar to why the Glm model performs worse on DouDizhu compared to Qwen and Llama (Line 240-244). The 14B model did not effectively balance the learning between the two roles and focused more on the landlord role, leading to weaker performance for the farmer role.
> | Model                | Landlord | Farmer | Avg. Win Rate |
> |----------------------|----------|--------|---------------|
> | Qwen2.5-7B-Instruct  | 0.828    | 0.784  | 0.806         |
> | Qwen2.5-14B-Instruct | 0.858    | 0.570  | 0.714         |
>
> ```
> Q3: Why do all the finetuned models obtain a score of 1.24 for Leduc in Table 2?
> ```
> As mentioned in Table 1, we used the official RLCard code to train a DQN model as the teacher model for Leduc. During testing, we found that the teacher model's performance on the validation set was also 1.24. Since Leduc has only six cards and two rounds of fixed-amount betting, we speculate that the teacher model learned some fixed strategies, and all three fine-tuned models have learned the strategies from the teacher.
>
> ```
> Q4: L121: Does a standard deck not have 52 cards?
> ```
> In DouDiZhu, a standard deck of 52 cards is used along with a Red Joker and a Black Joker. We will update the statement in Line 121.

---

> > ### Author Response · Authors · 2025-08-04
> > **Gentle Reminder from Authors**
> >
> > This is a gentle reminder that we are still awaiting your response regarding our rebuttal. We hope that our response has addressed your concerns and would like to hear from you. Please let us know if you have any further questions or feedback.

---

> > ### Comment · Reviewer_koKz · 2025-08-07
> >
> > I thank the authors for their response, which answers some of my questions.
> >
> > > "The poor initial performance is due to the base model's lack of strategic knowledge, or its inability to utilize existing strategic knowledge to make decisions in the environment."
> >
> > I don't understand the authors' persistence in providing an explanation for poor performance if they cannot provide any proof for such an explanation. The poor performance may be due to a lack of strategic knowledge, or it may be due to an inability to utilize strategic knowledge to make decisions, or it may be something completely different, or it may be a combination of the above. Unless there is evidence for one of these explanations, I would just state that the models perform poorly.
> >
> > > General capability encompasses various aspects and currently lacks a precise definition. Therefore, accurately measuring a model's general capability is relatively challenging. In our experiments, we investigated commonly used benchmarks from language model technical reports and selected typical benchmarks in three areas—knowledge-based Q&A, math, and code—as representatives to measure the model's general capability. In the future, we consider including other commonly used general capability evaluation benchmarks.
> >
> > While I agree that general capability lacks a precise definition, picking a fixed set of evals and (possibly) overfitting to those is clearly not general capability. I encourage the authors to consider including other commonly used evals (that they do not optimize for), as they suggest, if they want to make a claim about general capability.
> >
> > > Below, we have supplemented the performance of the model on games after training with general data (with results before training listed for comparison). From the table, it can be seen that the model's performance on games has remained mostly unchanged (slight improvements or stability in 5 games, and slight decreases in 3 games), indicating that the model has essentially regained a certain level of general capability while maintaining its gaming ability.
> >
> > Thank you for providing this additional experiment! It is very important and should be featured centrally in the paper. What is quite confusing to me is that training on "general-purpose" data enhances game-playing capabilities on most games. Do the authors have a hypothesis for this?
> >
> > > Each game can control the initial hand data by specifying a random seed. We used different random seeds during training and evaluation to ensure that the hand data differed between these two phases.
> >
> > Using different random seeds is better than not controlling for initial conditions, but I fear that the distributions might still have significant overlaps. Could you provide an analysis of how many initial hands appear in both the training and evaluation datasets? Ideally, they would be completely distinct.
> >
> > > As mentioned in Table 1, we used the official RLCard code to train a DQN model as the teacher model for Leduc. During testing, we found that the teacher model's performance on the validation set was also 1.24. Since Leduc has only six cards and two rounds of fixed-amount betting, we speculate that the teacher model learned some fixed strategies, and all three fine-tuned models have learned the strategies from the teacher.
> >
> > Even the models that were re-tuned on "general-purpose" data all achieve a score of 1.24. To me, this is highly suspicious and probably indicative of a bug.

---

> > > ### Author Response · Authors · 2025-08-09
> > >
> > > Thank you for your time and effort in reviewing our paper.
> > >
> > > As the discussion period is nearing its end with **less than one day remaining**, I wanted to ensure we have addressed all your concerns satisfactorily. If there are any additional points or feedback you'd like us to consider, please let us know. Your insights are invaluable to us, and we're eager to address any remaining issues to improve our work.

---

> ### Author Response · Authors · 2025-08-07
>
> Thank you very much for your response. We also greatly appreciate your valuable suggestions. Below, we will address the issues you raised. If you have any further questions, please feel free to ask us, and we will do our best to answer.
>
> ```
> I don't understand the authors' persistence in providing an explanation for poor performance. Unless there is evidence for one of these explanations, I would just state that the models perform poorly.
> ```
> Thank you for your suggestion. We will just state the models perform poorly and remove the explanation.
>
> ```
> While I agree that general capability lacks a precise definition, picking a fixed set of evals and (possibly) overfitting to those is clearly not general capability. I encourage the authors to consider including other commonly used evals (that they do not optimize for), as they suggest, if they want to make a claim about general capability.
> ```
> **We have supplemented the results of the models on four other common benchmarks (GQPA-Diamond, AIME2024, LiveCodeBench, IFEval) before and after fine-tuning on general mixed data.**
>
> | Model                             | MMLU-Pro | Math-500 | HumanEval | GQPA-Diamond | AIME2024 | LiveCodeBench | IFEval | Average_First_Three | Average_First_Six | Average_ALL |
> |-----------------------------------|----------|----------|-----------|--------------|----------|---------------|--------|---------------------|-------------------|-------------|
> | Llama-3.1-8B-Instruct             | 47.95    | 46.60    | 70.73     | 21.21        | 6.67     | 20.25         | 74.68  | 55.09               | 35.57             | 41.16       |
> | Llama-3.1-8B-Instruct-mix         | 44.74    | 35.20    | 60.98     | 26.77        | 6.67     | 17.75         | 74.31  | 46.97               | 32.02             | 38.06       |
> | Llama-3.1-8B-Instruct-mix-general | 45.18    | 47.20    | 65.24     | 27.27        | 10.00    | 13.50         | 68.95  | 52.54               | 34.73             | 39.62       |
> | GLM-4-9B-Chat                     | 48.04    | 52.40    | 76.22     | 26.26        | 0.00     | 18.00         | 69.13  | 58.89               | 36.82             | 41.44       |
> | GLM-4-9B-Chat-mix                 | 35.12    | 37.20    | 39.63     | 26.26        | 0.00     | 13.75         | 63.40  | 37.32               | 25.33             | 30.77       |
> | GLM-4-9B-Chat-mix-general         | 31.75    | 41.00    | 68.90     | 20.20        | 0.00     | 16.25         | 56.01  | 47.22               | 29.68             | 33.44       |
>
> From the table, it can be seen that if the general mixed data does not include a specific type of data, the model's corresponding capability will not be restored (after fine-tuning on the general mixed data, the performance of both models declined on the instruction-following benchmark).
>
> Based on your suggestion, we will also modify the expression related to general capability: by further fine-tuning on specific types of data, the model can restores its ability in specific areas to some extent, as demonstrated in the paper with general knowledge Q&A, mathematics, and coding capabilities.

---

> ### Author Response · Authors · 2025-08-07
>
> ```
> Thank you for providing this additional experiment! It is very important and should be featured centrally in the paper. What is quite confusing to me is that training on "general-purpose" data enhances game-playing capabilities on most games. Do the authors have a hypothesis for this?
> ```
> Thank you for acknowledging our response. Our hypothesis is that **the relative increase in the proportion of these games in the "general-purpose" data leads to the improvement in their performance.** The detailed explanation is as follows.
>
> After training on "general-purpose" data, the model's performance increased or remained stable mainly in five games, including Uno, Gin Rummy, Leduc, Limit, and Nolimit. Our paper uses two mixed datasets: the game mixed dataset and the general mixed dataset.
>
> The game mixed dataset contains 3.1 million data points, with the quantities for the eight games being: 700k, 950k, 650k, 200k, 50k, 250k, 200k, and 100k, respectively (lines 273-274). The general mixed dataset is composed of knowledge data, mathematics data, coding data, and game data. The proportions of these four types of data are 20k, 20k, 20k, and 8k, respectively. The 8k game data consists of 8 games, with 1k data points for each game (lines 335-338). To prevent forgetting game capabilities, we mixed data from 8 games into the general data.
>
> We trained on the game mixed dataset to obtain the mix model. Based on the mix model, we further trained on the general mixed dataset to obtain the mix-general model.
>
> From the composition of the two datasets, it can be seen that in the first dataset, the latter five games (Uno, Gin Rummy, Leduc, Limit, Nolimit) account for a relatively smaller proportion compared to the first two games (DouDizhu and GuanDan). Therefore, the mix model mainly shows improvement in DouDizhu and GuanDan (Figure 6).
>
> In the second dataset, the proportions of the 8 games are the same, meaning the data proportion of the latter five games is increased. Therefore, training on general-purpose data enhances or maintains game-playing capabilities in these five games.

---

> ### Author Response · Authors · 2025-08-07
>
> ```
> Using different random seeds is better than not controlling for initial conditions, but I fear that the distributions might still have significant overlaps. Could you provide an analysis of how many initial hands appear in both the training and evaluation datasets? Ideally, they would be completely distinct.
> ```
> We have tallied the number of initial hands that appear in both the training and evaluation datasets. The specifics are detailed in the table as described (the quantity during training for each game comes from Table 1 in the paper, and the quantity during testing is shown in lines 210-213 of the paper).
>
> | Game                   | # Training Games | # Test Games | # Initial Hands of Test in Training |
> |------------------------|------------------|--------------|-----------------------------|
> | DouDizhu               | 200,000          | 1,000        | 0                           |
> | GuanDan                | 6,000            | 20           | 0                           |
> | Riichi Mahjong         | 7,000            | 50           | 0                           |
> | Uno                    | 50,000           | 500          | 0                           |
> | Gin Rummy              | 50,000           | 100          | 0                           |
> | Leduc Hold’em          | 400,000          | 1,000        | 1,000                       |
> | Limit Texas Hold’em    | 200,000          | 1,000        | 7                           |
> | No-limit Texas Hold’em | 400,000          | 1,000        | 27                          |
>
> From the table, it can be seen that the  initial hands of test set for Leduc is completely present in the training set, while the training and test sets for the games Limit and Nolimit have a small overlap. Upon analysis, we found that Leduc uses only 6 cards and each player is dealt only one card at the start of the game. Considering the different player positions, there are only 30 unique initial hand combinations. The training set has completely covered these 30 combinations, which is why the initial hands of test set for Leduc is entirely present in the training set.
>
> We explain why we used 400,000 game records for training with Leduc. For the games Leduc, Limit, and Nolimit, we utilized the RLCard implementation [1]. In their paper [1], the authors trained on these three games for over 300,000 steps. Therefore, in this paper, we used a similar number of games for training these three games. Additionally, when playing these three games, community cards are dealt, and opponents using a random strategy often make random actions. These factors can all affect the game's state and increase its complexity. Therefore, **the number of games used for training should not be considered solely from the perspective of the initial hands.**
>
> Other examples include chess. In chess, the arrangement of pieces at the start of each game is the same, but the different moves by both sides cause the game's state to change dynamically, leading to diverse decision-making paths in each game. Therefore, training a strong chess AI often requires training on a large number of game trajectories. Furthermore, during game evaluation, the same game openings are still used.
>
> [1] Zha et al. RLCard: A Toolkit for Reinforcement Learning in Card Games
>
> ```
> Even the models that were re-tuned on "general-purpose" data all achieve a score of 1.24 on Leduc. To me, this is highly suspicious and probably indicative of a bug.
> ```
> As stated in our paper, the "general-purpose" data is composed of knowledge data, mathematics data, coding data, and game data, with proportions of 20k, 20k, 20k, and 8k data points respectively. The 8k game data consists of 8 games, with 1k data points for each game (lines 335-338). **Since the "general-purpose" data includes 1k of Leduc data (from the teacher model of Leduc)**, the models that were re-tuned on "general-purpose" data achieve a score of 1.24, which is the same as the score of the teacher model on Leduc.

---

### Official Review · Reviewer_PGiq · 2025-06-30

**Clarity:** 3
**Significance:** 2
**Originality:** 2
**Rating:** 3
**Confidence:** 3

**Summary:**

The paper conducts an empirical study to evaluate the capability of LLMs to handle card games. By giving the instruction prompt and supervisely fine-tuning LLMs with downstream datasets using LoRA, the paper shows that LLMs can master diverse card games and are able to simultaneously master multiple games by mixing trajectories together. Finetuning under a card game will hurt LLMs' general capabilities, while such decline can be marginally mitigated by integrating general instruction data.

**Questions:**

Descriptions of trajectory generation and the fine-tuning process are unclear; more details and a workflow figure might be more helpful.
In Figure 3, the authors pointed out that the reason for a low performance under Framer’s role in the DouDizhu game is due to the low quality of training data. While the assumption is reasonable, can the authors include further experiments to verify?
The authors talk about the influence between different games, starting from line 290, but did not specify the model they used for the experiment. All models should be tested to verify the conclusion.
The authors claimed that if the rules of two games are more similar, there tends to be greater knowledge transfer between them. However, while a model finetuned over GuanDan improved the DouDizhu’s performance, a model finetuned over DouDizhu still achieves a low performance on GuanDan. Does this imply that the transfer is not mutual?
The meaning of bolded values in tables should be mentioned in the caption.
A typo in line 162. Instead of Table 2, it should be Table 1.

I am mainly questioning the novelty and insights of this work. Whether the questions asked by the paper are valuable and whether the conclusion is general enough to be future development guidance.

**Ethical Concerns:**

["NO or VERY MINOR ethics concerns only"]

**Final Justification:**

The paper has some merit and responses indicate they can improve the current submission but the paper needs a resubmission with the revised results addressing some of the reviewer comments.

**Limitations:**

yes

**Quality:**

2

**Strengths And Weaknesses:**

The paper implemented extensive empirical studies, covered diverse card games, and multiple models with various model parameter scales. Some reasonable observations and conclusions come from experiments. However, I have several concerns:
The paper focuses on card games, which are less complicated than certain chess games like Go, and primarily evaluates the existing methods instead of proposing a new model or training strategy, which leads to a relatively narrow scope with limited insights.
The fine-tuning process conducted in the paper is solely using LoRa with fixed hyperparameters, making the paper less comprehensive. More fine-tuning strategies should be discussed.

---

> ### Author Rebuttal · Authors · 2025-07-31
>
> ```
> W1: The paper focuses on card games, which are less complicated than certain chess games like Go.
> ```
> The table lists the complexity of eight card games and Go[1][2]. From the table, it can be seen that **multiple games have a comparable complexity to Go (InfoSet Number). Additionally, unlike Go, these card games are games of incomplete information**, with a significant amount of hidden information in each state (InfoSet Size), thus presenting more uncertainties when making decisions.
> | Game                   | InfoSet Number | Avg. InfoSet Size |
> |------------------------|----------------|-------------------|
> | DouDizhu               | 10^53 ∼ 10^83  | 10^23             |
> | GuanDan                | 10^118         | 10^36             |
> | Riichi Mahjong         | 10^121         | 10^48             |
> | Uno                    | 10^163         | 10^10             |
> | Gin Rummy              | 10^52          | 10^9              |
> | Leduc Hold’em          | 10^2           | 10^2              |
> | Limit Texas Hold’em    | 10^14          | 10^3              |
> | No-limit Texas Hold’em | 10^162         | 10^3              |
> | Go                     | 10^171         | 0                 |
>
> [1] Zha et al. RLCard: A Toolkit for Reinforcement Learning in Card Games
>
> [2] Wikipedia. Game_complexity
>
> ```
> W2: I am mainly questioning the novelty and insights of this work. Whether the questions asked by the paper are valuable and whether the conclusion is general enough to be future development guidance.
> ```
> LLMs have achieved outstanding performance in a variety of tasks and are being applied to more tasks. Therefore, assessing the learning potential of LLMs on complex tasks is of significant importance.
>
> **The complexity of games can be measured relatively accurately, giving the conclusions of this paper a certain level of generality.** By evaluating the learning potential of LLMs in complex games, this paper can, to a certain extent, infer the learning potential of LLMs in tasks with specific complexities, providing guidance for deploying LLMs in a broader range of tasks.
>
> ```
> W3: The fine-tuning process conducted in the paper is solely using LoRA with fixed hyperparameters, making the paper less comprehensive.
> ```
> **Full parameter fine-tuning requires higher computational resources, so we adopted the widely used efficient fine-tuning method, LoRA.** This also facilitates other researchers in reproducing our work. LoRA fine-tuning has also demonstrated the powerful learning potential of LLMs. We acknowledge that, when resources are abundant, full parameter fine-tuning and more hyperparameter analysis experiments would be beneficial.
>
> ```
> Q1: In Figure 3, the authors pointed out that the reason for a low performance under Farmer’s role in the DouDizhu game is due to the low quality of training data. While the assumption is reasonable, can the authors include further experiments to verify?
> ```
> Currently, we manually check some data of Farmer’s role in DouDizhu to verify the existence of low-quality data. A more accurate approach would be to train a data quality classifier to filter out low-quality data before training the model and finally validate performance. Due to time and computational resource constraints, we will supplement this experiment in the future.
>
> ```
> Q2: The authors talk about the influence between different games, starting from line 290, but did not specify the model they used for the experiment. All models should be tested to verify the conclusion.
> ```
> Table 3 uses the Qwen model to analyze the influence between different games. Below are the influences between different games when using Llama and Glm. **The results of Llama and Glm show the same pattern as Qwen** (Line 292-299), where GuanDan has a positive influence on DouDizhu. Additionally, we can see that there are also positive influences among the three games: Leduc Hold’em, Limit Texas Hold’em, and No-limit Texas Hold’em.
>
> Llama: Influence between different games
> | Model     | DouDizhu    | GuanDan   | Riichi     | Uno         | Gin Rummy    | Leduc       | Limit      | Nolimit    |
> |-----------|-------------|-----------|------------|-------------|--------------|-------------|------------|------------|
> | DouDizhu  | **0.824**   | 0.000     | 0.13       | 0.008       | -0.496       | 0.637       | 1.14       | 3.21       |
> | GuanDan   | **_0.463_** | **0.598** | **_0.15_** | **_0.112_** | -0.390       | 0.637       | 0.88       | 0.96       |
> | Riichi    | 0.142       | 0.000     | **1.42**   | 0.060       | -0.242       | 0.757       | 0.95       | -1.07      |
> | Uno       | 0.234       | 0.000     | 0.04       | **0.160**   | **_-0.059_** | 0.637       | 1.14       | -0.472     |
> | Gin Rummy | 0.073       | 0.000     | 0.06       | **_0.112_** | **0.208**    | 0.637       | -0.19      | 3.08       |
> | Leduc     | 0.172       | 0.000     | 0.10       | 0.052       | -0.515       | **1.244**   | **_2.47_** | **6.98**   |
> | Limit     | 0.167       | 0.000     | 0.13       | 0.052       | -0.469       | **_1.105_** | **2.84**   | **_6.86_** |
> | Nolimit   | 0.170       | 0.000     | 0.04       | 0.056       | -0.198       | 1           | 2.06       | 4.92       |
> | Mix       | 0.870       | 0.661     | 1.38       | 0.164       | 0.186        | 1.244       | 2.77       | 6.02       |
>
> Glm: Influence between different games
> | Model     | DouDizhu    | GuanDan     | Riichi     | Uno         | Gin Rummy    | Leduc       | Limit      | Nolimit    |
> |-----------|-------------|-------------|------------|-------------|--------------|-------------|------------|------------|
> | DouDizhu  | **0.723**   | **_0.010_** | **_0.10_** | 0.060       | -0.460       | 0.637       | 1.14       | -1.21      |
> | GuanDan   | **_0.447_** | **0.629**   | 0.02       | **_0.136_** | -0.362       | -0.068      | -0.14      | 2.34       |
> | Riichi    | 0.063       | 0.000       | **1.33**   | 0.052       | **_-0.298_** | 0.282       | 0.50       | -5.15      |
> | Uno       | 0.111       | 0.000       | 0.06       | **0.176**   | -0.302       | 0.637       | 1.14       | 2.204      |
> | Gin Rummy | 0.075       | 0.000       | 0.06       | 0.016       | **0.196**    | 0.637       | 1.12       | 2.88       |
> | Leduc     | 0.142       | 0.000       | 0.08       | 0.048       | -0.416       | **1.244**   | **_2.41_** | **_6.02_** |
> | Limit     | 0.125       | 0.000       | 0.04       | 0.096       | -0.411       | **_1.232_** | **3.02**   | 5.05       |
> | Nolimit   | 0.114       | 0.000       | 0.06       | 0.004       | -0.499       | 0.648       | 1.53       | **6.24**   |
> | Mix       | 0.882       | 0.698       | 1.31       | 0.252       | 0.191        | 1.244       | 2.89       | 5.77       |
>
> Each row represents the performance of a model trained on one specific game across all games. Bold font indicates the maximum value, excluding the mixed training model. Italic font indicates the second-highest value, excluding the mixed training model.
>
> ```
> Q3: The authors claimed that if the rules of two games are more similar, there tends to be greater knowledge transfer between them. However, while a model finetuned over GuanDan improved the DouDizhu’s performance, a model finetuned over DouDizhu still achieves a low performance on GuanDan. Does this imply that the transfer is not mutual?
> ```
> From the results of the two games, DouDizhu and GuanDan, it can be seen that the transfer is one-way. However, from the results of the three games, Leduc Hold’em, Limit Texas Hold’em, and No-limit Texas Hold’em, the transfer is bidirectional. Compared to DouDizhu and GuanDan, the rules of these three games are more similar, leading to a more significant transfer effect.
>
> ```
> Q4: Descriptions of trajectory generation and the fine-tuning process are unclear; more details and a workflow figure might be more helpful. The meaning of bolded values in tables should be mentioned in the caption.
> ```
> Bold font in Table 2 indicates the maximum value in each category. Bold font in Table 3 indicates the two largest values in each category. We will provide more details about trajectory generation and the fine-tuning process, as well as more detailed table captions.

---

> > ### Author Response · Authors · 2025-08-04
> > **Gentle Reminder from Authors**
> >
> > This is a gentle reminder that we are still awaiting your response regarding our rebuttal. We hope that our response has addressed your concerns and would like to hear from you. Please let us know if you have any further questions or feedback.

---

> > > ### Comment · Reviewer_PGiq · 2025-08-04
> > > **response to the rebuttal**
> > >
> > > The authors provided more information to clarify the initial comments however, the paper needs a major revision thus I will hold my initial score.

---

> > > > ### Author Response · Authors · 2025-08-04
> > > >
> > > > Thank you very much for your feedback. We will update the supplementary content into the final paper. You can check if our reply has solved your problem. Or if you have any other questions, please feel free to ask us. Thank you.

---

### Note · Authors · 2025-08-11

We express our gratitude to all the reviewers for their valuable insights and for recognizing the contributions of our work. We are pleased that they have highlighted several key strengths of our paper, which we summarize below:

- **The reviewers praised the extensive and rigorous experimental design.** This includes the coverage of diverse card games, the use of multiple models with varying scales, detailed ablation studies. (Reviewer PGiq, Reviewer koKz, Reviewer Zt88, Reviewer  2Ka4)

- **The reviewers appreciated the in-depth analysis** that goes beyond just game performance to explore the correlation between fine-tuning data and performance, knowledge transfer between games, and the impact on general capabilities. (Reviewer Zt88, Reviewer 2Ka4)

- The research questions were found to be sensible, interesting, and forming a complete study. (Reviewer koKz)

- The work was valued for its practical implications, demonstrating that fine-tuned LLMs can achieve high performance in complex games. The paper was also credited for providing sufficient information for reproducibility. (Reviewer koKz, Reviewer 2Ka4)

- The paper was described as well-written, clear, easy to follow, and providing comprehensive analysis. (Reviewer koKz, Reviewer Zt88)


Below, we briefly summarize our work and the main additions to address all the reviewers' questions.

- Our work aims to evaluate the learning capability of language models through multiple complex card games. We conducted a comprehensive assessment focusing on mastering a single game, mastering multiple games, and recovering original specific abilities.

- In terms of mastering multiple games, we added evaluation results for reasoning models. We also added the influence between different games using Llama and Glm models.

- In terms of recovering original specific abilities, we supplemented the results of the model fine-tuned on general mixed data across eight games. We also added results on four other commonly used evaluation benchmarks.

Additionally, we have clarified and supplemented responses to many other questions raised by the reviewers. We will update our final paper with these revisions.

We want to emphasize that the evaluation of learning capabilities of language models is a valuable yet underexplored area, and our work contributes to the community by fostering a deeper understanding of learning capabilities of language models. We believe that it is worth publishing to stimulate further discussion.

---

### Decision · Program_Chairs · 2025-09-17

**Decision:**

Accept (poster)

**Comment:**

The paper investigates whether LLMs can master complex card games. The authors fine-tune multiple open-source LLMs on high-quality gameplay data across eight games, showing that fine-tuned LLMs achieve competitive performance. The study is comprehensive and provides open-source resources, which I believe are valuable.

The major concerns are that the novelty is limited, as the method mainly relies on SFT. However, given the extensive experiments and the released source code, I think the main contribution lies in its empirical insights and is acceptable. Another issue is that the claims about mastering multiple games are somewhat overstated, and the authors should carefully revise these claims in the final version. During the discussion, the authors provided further analyses and clarifications, which helped to convince some reviewers.

Overall, I find this work to be valuable in the community and recommend acceptance, with the expectation that the authors carefully refine their claims in the final version. In addition, since the main contribution of this paper is empirical, I strongly encourage the authors to release all data, code, and trained models.